# Hepatitis B Virus Genotypes and Subgenotypes Circulating in Belarus

**DOI:** 10.3390/cimb47060415

**Published:** 2025-06-04

**Authors:** Vladimir Eremin, Fedor Karpenko, Igor Karpov, Valery Semenov, Ina Oiestad

**Affiliations:** 1Republican Scientific and Practical Center for Transfusiology and Medical Biotechnology, 220053 Minsk, Belarus; fedor-doc@tut.by; 2Department of Infectious Diseases, Belarusian State Medical University, 220116 Minsk, Belarus; igorkarpov57@outlooc.com; 3Department of Infectious Diseases, Vitebsk State Medical University, 210009 Vitebsk, Belarus; vmsemenov@mail.ru (V.S.); inaoiestad@gmail.com (I.O.)

**Keywords:** HBV, genotypes/subgenotypes, sequencing, phylogenetic analysis

## Abstract

Approximately 800–900 new cases of chronic forms of hepatitis B are reported in Belarus annually. Compulsory vaccination, introduced to the country in the mid-90s, produces a certain positive effect on reducing the number of HBV cases. However, around 75,000 patients with chronic hepatitis B are estimated to live in the country at the moment. The main goal of this research was to determine the genotypes/subgenotypes of the hepatitis B virus and establish their role in maintaining the epidemic process in the country. Serological (CMIA, ELISA), molecular biological (PCR, sequencing), and bioinformatic (phylogenetic analysis) methods have been used to obtain results. As studies have shown, in 722 (81.7%) samples, genotype D has been determined; in 156 (17.7%), genotype A; in 3 (0.3%), genotype C; and only in 1 sample (0.1%), genotype B. In two cases (0.2%), a recombinant form of the hepatitis B virus has been detected. The epidemic process of hepatitis B in the country is supported mainly by the circulation of “local” variants of the virus. At the same time, there are occasional introductions of new genetic variants from outside of the country.

## 1. Introduction

Hepatitis B virus (HBV) infection remains a serious health concern despite the HBV vaccine being introduced in the late 1990s. Approximately two billion cases of the disease were reported worldwide from the time of HBV discovery to the beginning of the 21st century. More than 400 million of them subsequently became chronic HBV carriers [1]. Chronic HBV infection is considered a major cause of many liver-related medical complications such as hepatocellular carcinoma, liver cirrhosis, and liver failure [2,3,4]. Hepatitis B virus, like HIV, has three main transmission mechanisms: parenteral (through blood and its products and/or co-injection of drugs), vertical (from infected mother to child), and sexual transmission. HBV belongs to the Hepadnaviridae family, with a genome of approximately 3.2 kb in length. Due to the virus’s complexity, the HBV genome contains an overlapping region that encodes four different genes (S, C, P, and X). The S gene encodes exclusively the HBV surface envelope protein in its long open reading frame [5]. The presence of start codons allows the gene splicing into small, medium and large polypeptides, which are typically used either as a single target gene or in combination with other HBV genes to detect HBV DNA [5]. HBV core antigen is commonly encoded by the C gene, whereas the X gene encodes protein X [6]. The P gene encodes a polymerase protein, an integral component of the reverse transcription process during HBV replication. The HBV replication cycle lacks proofreading properties, resulting in progeny with a high genomic variability [7].

Due to significant genetic diversity, HBV is categorized into ten genotypes (A–J) with 7.5% intergroup variation [8]. In addition to E and G, all genotypes are classified into 25 subgenotypes (A1–A3, B1–B5, C1–C6, D1–D6, and F1–F4) with 4% amino acid variability [9,10]. HBV genotypes are distributed differently depending on the geography: HBV-B, HBV-C, and HBV-E are most common in Oceania and East Asia, while HBV-E is most prevalent in Central and West Africa. HBV-F and HBV-H are found only in Alaska and Latin America. In contrast, HBV-D is a global pandemic. In Australia, Europe, Indonesia, North Africa, and Western Asia, HBV-D1 is the most common virus, while HBV-D2 is found in Albania, Japan, Malaysia, Northeastern Europe, Russia, and the UK [11,12,13,14,15,16]. A systematic genotype and subgenotype re-ranking of hepatitis B virus under a novel classification standard [16].

The progression and natural course of the disease vary among different HBV genotypes. These factors can complicate HBV infection treatment since the efficacy of known therapeutics turns out to be ineffective against certain genotypes and new genotypic variants [8,9]. Thus, there is a global exigency for constant updates in genotypic information and surveys of HBV-infected patients [15].

In Belarus, selective hepatitis B vaccination for epidemic indications has been organized since 1996 and included in the national preventive vaccination calendar since 2000. Vaccination of newborns, children aged 13 years, and certain risk groups (health care workers and persons in domestic contact with infected persons) has helped to reduce the incidence of hepatitis B infection. Vaccination against viral hepatitis B has made it possible to reduce the incidence of acute hepatitis B almost six-fold over the past 10 years (from 5.9 to 1.02 per 100,000 population) and to consider Belarus as a country with a low incidence of acute viral hepatitis B (less than 2% of the population). This means that vaccination coverage against hepatitis B has been over 97 percent for several decades. This is a good result. In 2022, a WHO-supported study of the child population in Belarus showed how many of those vaccinated had HBsAg, the surface antigen of viral hepatitis B—it was detected in one child only. The most affected age groups are teenagers and adults 15 to 40 years old. The incidence rate in this specific age group is significantly higher than in other age groups of the population of the Republic. Approximately 800–900 new cases of chronic viral hepatitis B are detected annually in the country, despite a significant decrease in the occurrence of acute viral hepatitis B. According to the epidemiological essays, 75,000 patients with chronic hepatitis B currently live in Belarus.

## 2. Materials and Methods

**Samples.** In this study, 884 blood serum/plasma samples were tested for HBV markers with CMIA/ELISA and PCR methods. The samples were obtained from patients and blood donors with acute and chronic HBV from different regions of the country. The age of the patients ranged from 4 years to 78 years. Furthermore, 428 samples were obtained from males and 374 from females. In 82 cases, gender and age were not specified.

The results of the epidemiologic investigation showed that out of 884 patient samples, 367 (41.5%) were infected through sexual contact, 211 (23.9%) indicated medical manipulation, 49 (5.5%) were children born to infected mothers, 36 (4.1%) were injecting drug users, 21 (2.4%) indicated household contacts, 18 (2.0%) indicated non-medical manipulation (tattoos, etc.), and 182 (20.60%) patients were unable to specify any route of infection. Serum/plasma samples were collected and analyzed during 2013–2024.

**Serological tests.** The commercial kit chemiluminescent microparticle immunoassay (CMIA) Architect HBsAg Qualitative II Reagent Kit (Abbott, Lake County, IL, USA) was applied to detect HBsAg. Positive results were confirmed using the Architect HBsAg Qualitative II Confirmatory Reagent Kit (Abbott, USA), as well as ELISA “Vectogep B-HBs antigen”, manufactured by “Vector-Best”, Novosibirsk, Russia.

**PCR.** The polymerase chain reaction was used to determine the DNA of the hepatitis B virus using test systems produced by Vector-Best: “RealBest HBV DNA quantitative” and “RealBest HIV RNA quantitative” in accordance with the manufacturer’s instructions. Viral load in the samples ranged from 1.2 × 10^2^ to 3.4 × 10^6^ IU/mL HBV DNA.

**DNA isolation.** Viral RNA/DNA from the blood serum/plasma samples were isolated using the “Kit of reagents for the isolation of NK” (manufactured by “Vector-Best”, Russia) and the “Kit of reagents for the isolation of RNA/DNA from clinical material” “RIBO-sorb”, “RNA-prep”, Moscow, Russia. The manufacturers’ instructions were strictly followed.

**Nested PCR** was performed using previously described primer pairs p1/pR5 (p1: 5-CCTGCTGGTGGCTCCAGTTC-3 at nucleotide position 55—6, and primer pR5: 5-GGT TGC GTC AGC AAA CAC TTG-3 at position 1197—1178) and p4/pR2 (p4 5-CTC ACA ATA CCG CAG AGT CTA GAC T-3 at nucleotide position 230—254, and pR2: 5-AAA GCC CAA AAG ACC CAC AAT-3 at position 1017—997) [17] according to the following protocol in a volume of 25 µL: 2.5 µL 10× buffer + MgCl_2_; 0.25 µL 25 mM dNTPs; 0.5 µL 10 µM p1; 0.5 µL 10 µM pR5; 0.8 µL 10 µM Taq polymerase; and 18.45 µL bdH_2_O. The first round of PCR was carried out according to the following protocol: one denaturation cycle at 95 °C for 2 min and 30 cycles of amplification were performed with denaturation at 95 °C for 30 s, annealing at 53 °C for 30 s, and extension at 68 °C for 1 min, then one cycle at 68 °C for 5 min and storage at 10 °C. In the second round of PCR, the annealing temperature was 50 °C; the rest of the parameters were not changed. As a result, we obtained a fragment of about 900 base pairs.

**Amplified DNA** fragments were analyzed on a 2% agarose gel (Condalab, Madrid, Spain). Electrophoresis was carried out at 10 w/cm of gel in TRIS-acetate buffer, pH 8.0. DNA was visualized using the Vitran Photo gel documentation system (Biocom Company LLC, Stavropol, Russia). The fragment size was determined according to the molecular weight of a marker of 100–1000 bp (Fermentas, Vilnius, Lithuania).

The resulting DNA fragments were purified using the NimaGen ExS-Pure kit, Nijmegen, The Netherlands.

**PCR sequencing** was carried out using the second-round primer pair p4/pR2 in the volume of 10 µL according to the following protocol: 5× seqbuf-2 µL; BigDye Terminator v. 3.1 (Applied Biosystems)—1 µL; p4/pR4—2 µL; bdH_2_O—3 bdH_2_O; and DNA—2 µL.

**Electrophoresis of HBV DNA fragments** obtained and purified after PCR sequencing was performed on an AB 3500 genetic analyzer, USA.

**Phylogenetic analysis** of the obtained HBV DNA fragments was conducted using the computer programs Sequencing Analysis v.6, BioEdit, SeqScape v3, MEGAX, and Genious v.8.1. Phylogenetic trees were built using the ML (maximum likelihood) algorithm in the PHYML V.3.0 program. The SH-aLRT test was performed to calculate the statistical significance of the clusters. Clusters with a support node ≥ 0.9 were considered reliable.

**Mutations in the HBV genome** in the S and P regions were determined using the following programs: https://www.geno2pheno.org, http://www.hiv-grade.de/hbv_grade/deployed/grade.pl?program=hbvalg, and https://hivdb.stanford.edu/HBV/HBVseq, accessed on 22 October 2024.

## 3. Results

Our studies showed the following: 722 (81.7%) out of all 884 sequenced and analyzed samples had genotype D; 156 (17.7%)—Genotype A; 3 (0.3%)—Genotype C; and only 1 (0.1%)—Genotype B. For two cases (0.2%), a recombinant form of the hepatitis B virus was detected. Genotype D was found to be represented by subgenotypes D1 (145/20.1%, ±3.26), D2 (371/51.4%, ±5.3), D3 (201/27, 8%, ±3.5), and D4 (5/0.7%, ±0.3); A-A2 (156/17.7%, ±2.9); C1-1 (0.11%) and C2-2 (0.23%); B-B4 (0.11%); and recombinant forms C2/D (0.11%) and A2/C2 (0.11%) were detected (Figure 1).

In 12 cases, viruses with mutations to reverse transcriptase inhibitors Lamivudine, Zeffix^®^, Telbivudine, Tyzeka^®^, and Sebivo^®^ and partial resistance to Entecavir and Baraclude^®^ were identified. Moreover, in four cases, resistance mutations were identified in subgenotypes A2 and D2; in three cases—in the genome of subgenotype D1; and in one case—on subgenotype D3. Mutations were most often recorded at positions 180 M and 204 V—in eight cases, 204 I—in three cases, 80 I and 173 L—two times each, and 80 V and 181 T in one case (Table 1).

Determining mutations is very important for infectious disease doctors, since substitutions at positions 180 M, 181 T, 204 V, and 204 I lead to resistance to Lamivudine (Zeffix), Adefovir (Hepsera), and Telbivudine (Tyzeka, Sebivo) and require a change in treatment regimen.

Eleven clusters were identified while analyzing samples of subgenotype D1, Figure 2. Five large clusters (1–5) containing 14 or more HBV DNA sequences, as well as small clusters 9–11, contained samples of HBV detected in citizens permanently residing in Belarus. Clusters 2, 4, and 6–8 contained HBV DNA sequences detected in patients from Ukraine (cluster 4), Pakistan (cluster 6), Turkmenistan and Afghanistan (clusters 2 and 7), and Azerbaijan (cluster 8).

Overall, the samples, as can be seen in Figure 2, were not related to each other, as the support node was less than 0.9. At the same time, in cluster 5, samples MS_148_D1_Vit and MS_149_D1_Vit had a support node of 0.976, indicating a possibly single source of origin of the virus in the patients. Indeed, the samples were from a mother and her vertically infected child.

The D2 subgenotype consisted of nine clusters, and it should be noted that more than 80% of all HBV DNA sequences were concentrated in clusters 1–3, which contained 35 or more sequences (Figure 3).

In this study, 20% of the remaining samples were concentrated in the remaining six clusters. It is important to note that the D2 subgenotype was exclusively formed by samples obtained from patients permanently residing in Belarus. Most of the samples were unrelated and had a support node less than 0.9.

HBV DNA sequences of subgenotype D3 formed eight clusters and consisted mainly of samples obtained from patients living permanently in Belarus (Figure 4). The first four clusters contained almost 90% of all samples. The seventh cluster was formed by samples obtained from patients from one family (a mother and two children) who came to Belarus for treatment from the Kyrgyz Republic.

HBV DNA sequences of subgenotype A2 formed at least six clusters, with the first three accounting for almost 90% of all samples (Figure 5). All analyzed sequences were obtained from patients permanently residing in Belarus.

All five HBV DNA sequences of subgenotype D4 were obtained from epidemiologically unrelated patients from Belarus. Samples of subgenotypes C1 and C2 were identified in adult patients and a child from China. The subtype B4 sample was revealed in a student from Vietnam who came to study in our country.

Finally, both samples with recombinant forms of HBV were identified in residents of Belarus.

## 4. Discussion

Approximately 800–900 new cases of chronic forms of hepatitis B are reported in Belarus annually. Compulsory vaccination, introduced to the country in the mid-90s, produces a certain positive effect on reducing the number of HBV cases. However, around 75,000 patients with chronic hepatitis B are estimated to live in the country at the moment.

Our previous study showed that subgenotype D2 (55.8%) was dominant in Belarus, followed by D3 (18.3%), D1 (11.6%), and A2 (11.6%) [18]. The conducted studies show that the structure of genotypes/subgenotypes in the country has actually remained unchanged. Genotype D still dominates in the country and accounts for 81.7% of all analyzed cases. Genotype D was found to be represented by subgenotypes D1 (145/20.1%), D2 (371/51.4%), D3 (201/27.8%), and D4 (5/0.7%); A-A2; C-C1 and C2; B-B4; and recombinant forms C2/D and A2/C2 were detected. Possible causes of the infection in patients were determined, i.e., they occurred due to the circulation of “domestic” variants of the virus and/or the introduction of the virus from outside. The findings obtained show that the epidemic transmission of viral hepatitis B in the country is maintained mainly due to the circulation of local variants of the virus. Subgenotypes C1 and C2, as well as B4, were identified in patients from China and Vietnam who came to Belarus to study and work. Two recombinant forms, C2/D and A2/D, were identified in the residents of our country.

The large number of clusters identified in groups of different subgenotypes may be due to repeated introductions of the virus into the country. This occurred both in pre-vaccine times and at present, as evidenced by samples from Asian countries.

Comparing the distribution of different HBV genotypes in neighboring countries and even in different regions of the same country, we may conclude that they vary. For example, genotype D significantly dominates in the European part of the Russian Federation, while, for example, in Yakutia, genotype A occupies one of the leading positions (36.4% A and 58.6% D) [19]. In Asia, particularly in its eastern part, genotypes B and C dominate, and the C2 subgenotype is epidemic in China [20,21]. In Western Europe, in particular in Portugal, HBV genotype A (HBV/A) was the most prevalent genotype (41.5%), followed by D [HBV/D; (33.8%)] and E [HBV/E; (24.6%)]. Subgenotypes A1 (HBV/A1) and D4 (HBV/D4) were almost equally prevalent, with 23.1% and 22.3%, respectively, followed by A2 (HBV/A2) with 16.2% and D3 (HBV/D3) with 11.5% [22].

On the African continent, genotype E was most represented and significantly outnumbered all other genotypes. By region, genotype A had the highest cumulative prevalence in eastern and southern Africa, E in western Africa, and D in northern Africa (*p* < 0.0001). Regarding the new genotypes B and C on the African continent, genotype B was significantly higher in southern Africa than C (*p* < 0.001). In contrast, genotype C was significantly higher in eastern Africa than in western Africa (*p* < 0.0001). A1 and D/E were the most diverse subgenotypes and mixtures of genotypes, respectively [23].

According to the European Centre for Disease Prevention and Control, 34 (2.6 per 100,000 population) cases of viral hepatitis B were registered in Estonia in 2022, of which 2 were acute (0.2 per 100,000 population) and 32 chronic (2.4 per 100,000 population). In neighboring Latvia during the same period, 191 (10.2 per 100,000 population) cases of viral hepatitis B were registered, of which 17 (0.9 per 100,000 population) were acute cases and 174 (9.3 per 100,000 population) were chronic viral (1.0 per 100,000 population) cases of viral hepatitis B, of which 7 (0.2 per 100,000 population) were acute and 20 (0.7 per 100,000 population) were chronic viral hepatitis B cases [24].

The Republic of Belarus is geographically located in the center of Europe. The residents of the country are actively traveling, and many people from other countries relocate to Belarus for work and study. This fact can easily explain the introduction of the new variants of HBV to the Republic.

The results of this study contribute valuable information on the molecular genetic characteristics of HBV prevailing in the Republic of Belarus. Possible routes of the introduction of the new variants of HBV have been identified. The main mutations in the P region of the virus genome leading to resistance to antiviral drugs have been detected. Identification of the mutations at positions 181 T, 204 V, and 204 I, which determine resistance not only to Lamivudine (Zefffix) but also to Adefovir (Hepsera) and Telbivudine (Tyzeka and Sebivo), made it possible to use Tenofovir DF for treatment of the patients. The compensatory mutations 80 V, 173 L, and 80 I have been found in the complex with the mutations leading to virus resistance to antiviral drugs.

## 5. Conclusions

Summarizing the results obtained, we can once again note that the epidemic process of viral hepatitis B in Belarus is maintained mainly due to the circulation of local variants of the virus. Even in spite of vaccine prophylaxis, about 1000 new cases of chronic viral hepatitis B are detected annually in the country. The detection of new cases, for example, by blood donor testing, may indirectly confirm this. At the same time, as our results show, new variants of hepatitis B virus are entering the country from outside the country with workers and students, for example, from Southeast Asian countries, as well as with citizens of our country who worked outside Belarus. Currently, these subgenotypes of hepatitis B virus are not involved in the epidemiologic process and are detected in individual patients, but their active inclusion in the epidemiologic process cannot be excluded in the future.

## Figures and Tables

**Figure 1 cimb-47-00415-f001:**
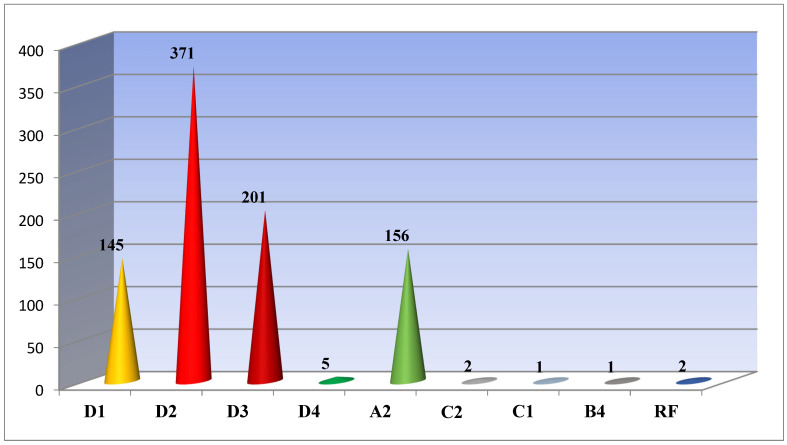
HBV subgenotypes identified in Belarus.

**Figure 2 cimb-47-00415-f002:**
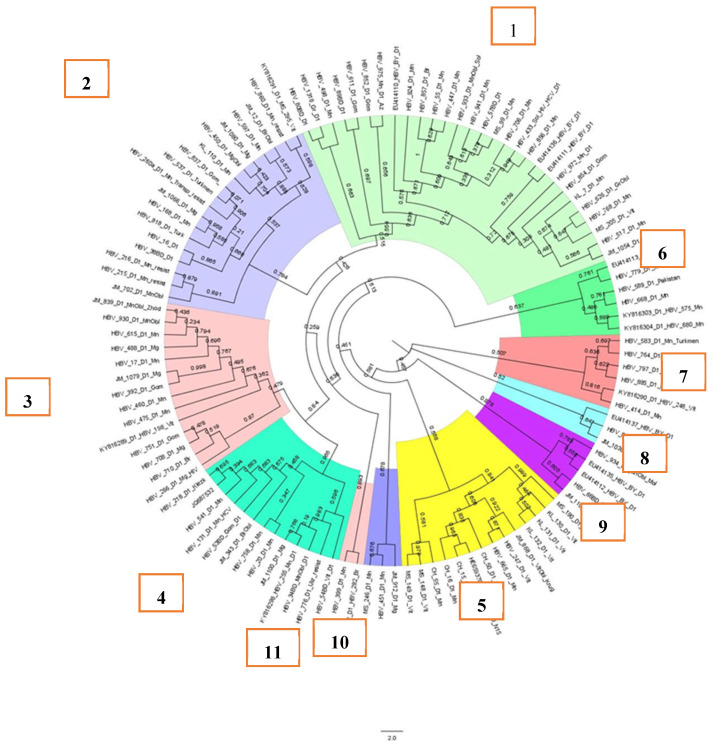
Phylogenetic analysis of HBV subgenotype D1.

**Figure 3 cimb-47-00415-f003:**
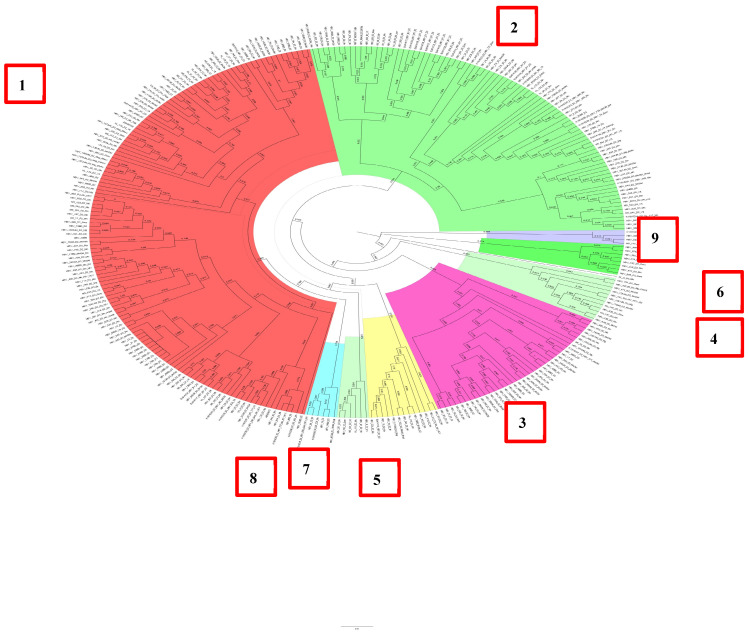
Phylogenetic analysis of HBV subgenotype D2.

**Figure 4 cimb-47-00415-f004:**
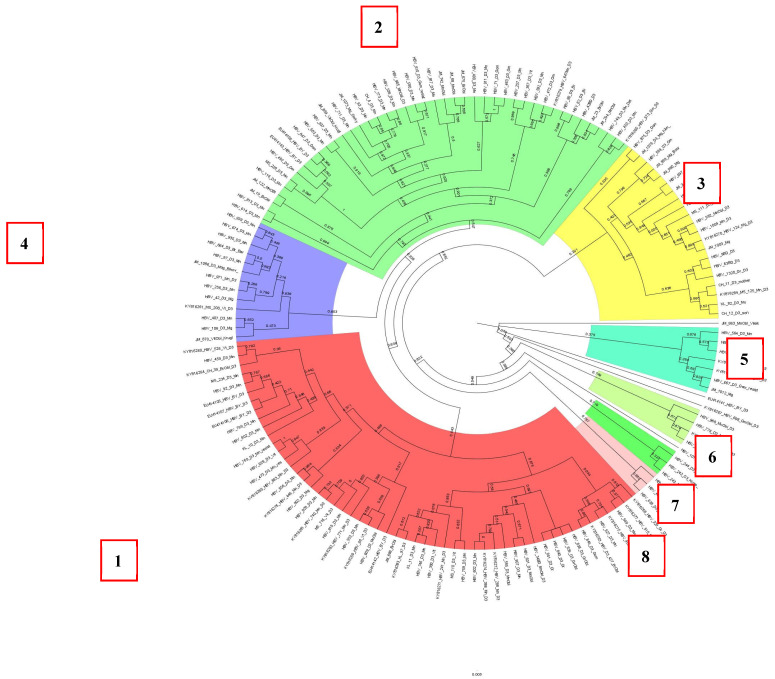
Phylogenetic analysis of HBV subgenotype D3.

**Figure 5 cimb-47-00415-f005:**
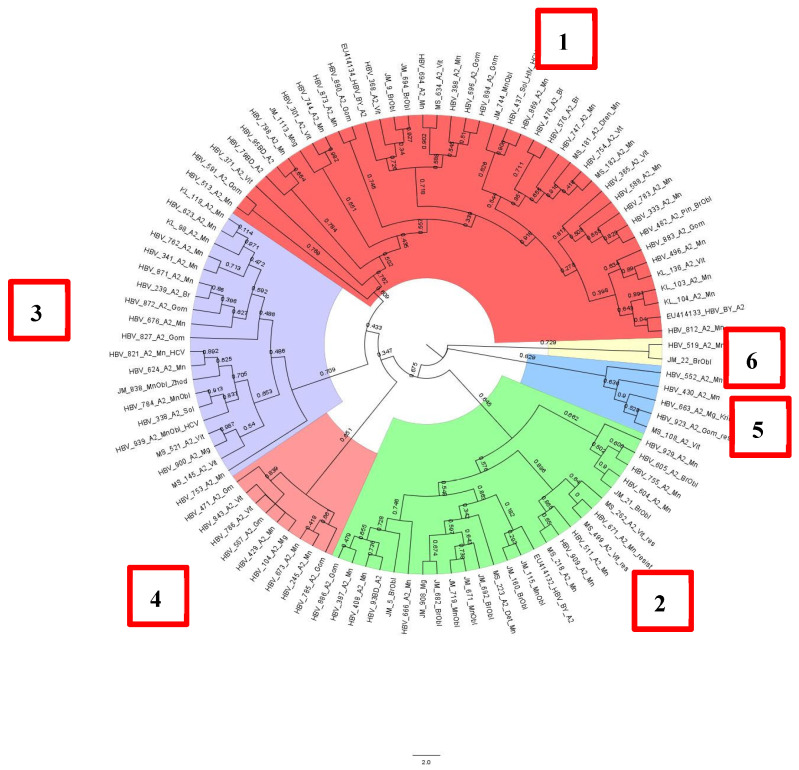
Phylogenetic analysis of HBV subgenotype A2.

**Table 1 cimb-47-00415-t001:** Resistance of mutations identified in different HBV subgenotypes.

N	Subgenotypes	Mutations
1	A2	80 V−1; **180 M**-3; **181 T**-1; **204 V**-3;
2	D1	173 L-1; **180 M**-3; **204 V**-3;
3	D2	80 I-2; 173 L-1; **180 M**-2; **204 V**-2; **204 I**-2;
4	D3	**204 I-**1;

## Data Availability

Data are contained within the article.

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
