# Peer review of "Hepatitis B Virus Genotypes and Subgenotypes Circulating in Belarus"

_cimb, 2025, doi:10.3390/cimb47060415_

Round 1
Reviewer 1 Report (New Reviewer)
Comments and Suggestions for Authors
This manuscript presents an investigation into the genotypic characteristics of the Hepatitis B virus in Belarus, offering valuable insights into the epidemiology of HBV infection within this region. The cluster analysis of various subtypes, which hints at possible sources of infection, is of particular interest. Nonetheless, several aspects of the study warrant further enhancement.
-
The paper currently lacks any analysis or description of the G145A/R mutation, which is associated with resistance to vaccination. It is imperative that this element be addressed in future revisions.
-
Although the authors have conducted cluster analyses on multiple subtypes, there is a noticeable absence of discussion regarding the relationship between these clusters and drug resistance mutations. It is recommended that this analysis be expanded to include such correlations.
-
While the investigation acknowledges the presence of mutations indicative of HBV resistance to antiviral drugs, there is no mention of the usage patterns of lamivudine and entecavir. Detailed data on the specific frequencies of usage for these drugs should be incorporated to provide a more comprehensive understanding of the resistance mechanisms.
-
The hypothesis that the virus has been introduced by international students and subsequently recombined with local strains is posited without substantial evidence. To strengthen the credibility of this claim, it is essential that additional analyses be provided to substantiate this assertion.
By addressing these points, the study can significantly enhance its contribution to the understanding of HBV dynamics in the Belarusian context.
Comments on the Quality of English LanguageThere are no specific comments regarding the quality of the English used in the manuscript.
Author Response
Dear Colleague,
Thank you for your questions and recommendations. I will try to answer them in as much detail as possible.
Question: The paper currently lacks any analysis or description of the G145A/R mutation, which is associated with resistance to vaccination. It is imperative that this element be addressed in future revisions.
Answer: Yes, indeed, the study of “vaccine-escape” mutations is an important area of research, given the rather high level of variability in hepatitis B virus, both due to the presence of the reverse transcriptase enzyme and under the influence of antivirals and vaccines. The G145A/R mutation described by Prof. Zanetti A.R. et al. is the most common “vaccine-escape” mutation. Our studies did not detect this mutation, but other “vaccine-escape” mutations were identified:120P, 120T,126I, 129R, 128V, 131N, 130D,144A, 144E. A description of the distribution of these mutations and mutations associated with occult HBV (OBI) is anticipated in our next publications.
Question: Although the authors have conducted cluster analyses on multiple subtypes, there is a noticeable absence of discussion regarding the relationship between these clusters and drug resistance mutations. It is recommended that this analysis be expanded to include such correlations.
Answer: In this paper we wanted to present in general the molecular epidemiologic situation on viral hepatitis B in our country and try to find changes compared to 2002-2007 when the first large paper on this topic was published. Out of such a large cluster of genotypes, only four samples had antiviral resistance mutations. These mutations are summarized in Table 1 in the article. These mutations and patients will be described by us in more detail in the next publication.
Question: While the investigation acknowledges the presence of mutations indicative of HBV resistance to antiviral drugs, there is no mention of the usage patterns of lamivudine and entecavir. Detailed data on the specific frequencies of usage for these drugs should be incorporated to provide a more comprehensive understanding of the resistance mechanisms.
Answer: In this paper we do not present treatment regimens for patients with chronic hepatitis B, this will be described in our next publication. However, we can say that since the D genotype of the virus is dominant in our country, these patients were on pegylated interferon alpha 2a+lamivudine.
Question: The hypothesis that the virus has been introduced by international students and subsequently recombined with local strains is posited without substantial evidence. To strengthen the credibility of this claim, it is essential that additional analyses be provided to substantiate this assertion.
Answer: We are in no way asserting or hypothesizing the predominant role of students coming to our country to study in the spread of hepatitis B. The article only mentions a case with the B4 subgenotype. The Republic of Belarus is geographically located in the center of Europe. The residents of the country are actively travelling and many people from other countries relocate to Belarus for work and study. This fact can easily explain the introduction of the new variants of HBV to the Republic. We have a broad cluster analysis using reference sequences from GenBank, where phylogenetic relationships of virus sequences from Belarus with sequences from other countries and continents are presented. Unfortunately, these phylogenetic trees are very large and cannot be presented in the paper. However, we will try to prepare a separate publication where links between HBV from Belarus and the virus from other countries will be established.
Reviewer 2 Report (New Reviewer)
Comments and Suggestions for Authors
The authors have investigated the presence of B Virus Genotypes in the European state of Belarus.
- Please explain clearly why the target population was in Belarus. What was the significant differences versus populations in neighbouring countries? Did you collaborate with researchers in neighbouring countries to make common study?
- Please define clearly the objectives of the study. Please explain the gaps in the international literature that would be filled from this work. Please explain the differences of this study in comparison to previous relevant works.
- Please add a paragraph with review of the situation in north-east Europe.
- Please describe the novelty of this work at international level and justify the performance of the study.
- Please add a new sub-section in M&M to describe in great clarity all the control procedures and material employed in this study.
- Please include names of manufacturers of all consumables and equipment used.
- The visualization of the manuscript is excellent. However, please increase use of tables to present the results and also reduce the relevant text.
- There are some very recent references on the topic that are missing. Please include and please compare the findings to those of the present study.
- The conclusions are a bit over-optimistic. Please tone down the section to bring in line with the findings and please also explain the future directions of this research.
Overall. Useful study. Extensive changes and re-evaluation after correction.
Author Response
Dear colleague,
thank you for your questions, I will try to answer them in detail.
Question: Please explain clearly why the target population was in Belarus. What was the significant differences versus populations in neighbouring countries? Did you collaborate with researchers in neighbouring countries to make common study?
The answer: The results presented in this article were obtained in Belarus. The target group is patients with chronic form of viral hepatitis B, permanently residing in the territory of our country. Individual serum/blood plasma samples were from patients from other countries who came to work or study in Belarus.
Question: Please define clearly the objectives of the study. Please explain the gaps in the international literature that would be filled from this work. Please explain the differences of this study in comparison to previous relevant works.
The answer: The present work differs from our previous article (2008) in that it is based on a larger volume of material and reflects the molecular epidemiologic situation on HBV in the country at the present stage. The main goal of the study was to provide a molecular-epidemiological characterization of hepatitis B virus circulating in Belarus.
Question: Please add a paragraph with review of the situation in north-east Europe.
The answer: Thank you for the suggestion. We will definitely add a fragment with the situation of viral hepatitis B in the north east Europe.
Question: Please describe the novelty of this work at international level and justify the performance of the study.
The answer: In this publication, for the first time, the molecular epidemiologic situation on viral hepatitis B at the present stage is given on a large volume of material. The work can be said to partially supplement our earlier studies (2008), conducted on a small amount of material.
Question: Please add a new sub-section in M&M to describe in great clarity all the control procedures and material employed in this study.
The answer: May I clarify which control procedures and materials you are talking about? All procedures and materials are described in detail in the materials and methods section.
Question: Please include names of manufacturers of all consumables and equipment used.
The answer: We have listed the manufacturer of some of the materials.
Question: The visualization of the manuscript is excellent. However, please increase use of tables to present the results and also reduce the relevant text.
The answer: Thanks for the offer. However, we intentionally left only the figures with phylogenetic trees, which allow us to assess in general the molecular epidemiology of hepatitis B in our country. These trees show that the virus was introduced into the country many times and of different subgenotypes.
Question: There are some very recent references on the topic that are missing. Please include and please compare the findings to those of the present study.
The answer: We supplemented the literature with references on the epidemiology of viral hepatitis B in the Baltic region and in Africa. For a broader view of the distribution of different virus genotypes and the epidemiology of hepatitis B in different regions of the world.
Question: The conclusions are a bit over-optimistic. Please tone down the section to bring in line with the findings and please also explain the future directions of this research.
The answer: Thanks for the tip, we have corrected the final part of the article. Thank you.
Reviewer 3 Report (New Reviewer)
Comments and Suggestions for Authors
The focus of your study is to determine the genotypes and subgenotypes of the hepatitis B virus in Belarus.
The practical problem currently in Belarus is that about 1000 new cases of chronic hepatitis B are detected annually. Since these cases are often occurring as a result of circulation of local variants, it might be worthwhile to "test and treat". Specifically test the blood sample while they wait and start them on treatment before they leave.
Author Response
Dear colleague,
thank you for your questions, I will try to answer them in detail.
Question: The practical problem currently in Belarus is that about 1000 new cases of chronic hepatitis B are detected annually. Since these cases are often occurring as a result of circulation of local variants, it might be worthwhile to "test and treat". Specifically test the blood sample while they wait and start them on treatment before they leave.
The answer: In practice, it turns out that first we have to do all the tests: general blood test, biochemistry, determine the viral load, determine the genotype of the virus and some other tests. When the doctor has all the results in hand, he prescribes treatment for the patient.
Round 2
Reviewer 2 Report (New Reviewer)
Comments and Suggestions for Authors
All the issues were addressed. No further comments.
This manuscript is a resubmission of an earlier submission. The following is a list of the peer review reports and author responses from that submission.
Round 1
Reviewer 1 Report
Comments and Suggestions for Authors
In the present study, the authors are interested in the status of HBV infection in Belarus. Specifically, they analyzed a total of 884 HBV-positive samples and determined the molecular epidemiology of circulating HBV strains in Belarus during 2014 to 2023. Overall, the topic is novel; however, the results are not well presented. The manuscript needs to be greatly improved before publication.
1. The Abstract section is incomplete, which only contains results. Please include a few sentences of introduction, methods and materials, and conclusion.
2. Introduction section: Background information of HBV infection in Belarus is needed.
3. Materials and Methods section: Consider dividing the whole section into several parts with subtitles for a better readability.
4. Results section: Figures 2 to 12 are hard to comprehend. Please specify the red labels in each figure legend. Additionally, the letters are too small to read.
5. In the phylogenetic trees, the numbers in the branches are unnecessary to be shown. Some representative ones should be enough.
6. The authors should upload all relevant HBV sequences derived from this study to the GenBank database.
7. Discussion section: I wonder why there is no discussion of HBV mutations since you performed mutational analysis.
8. Please standardize the abbreviation of hepatitis B virus as HBV throughout the manuscript.
Comments on the Quality of English LanguageProofreading of the manuscript by an English native speaker is recommended.
Author Response
HEPATITIS B VIRUS GENOTYPES AND SUBGENOTYPES CIRCULATING IN BELARUS
1Vladimir Eremin, 1Fedor Karpenko, 2Igor Karpov, 3Valeriy Semenov, 3Ina Oiestad
1Republican Scientific and Practical Center for Transfusiology and Medical Biotechnology
2Belarusian State Medical University, Department of Infectious Diseases, Belarus
3Vitebsk State Medical University, Department of Infectious Diseases, Belarus
Abstract. Approximately 800-900 new cases of chronic forms of hepatitis B are reported in Belarus annually. Compulsory vaccination, introduced to the country in the mid-90s, produces a certain positive effect on reducing the number of HBV cases. However, around 75,000 patients with chronic hepatitis B are estimated to live in the country at the moment. The main goal of this research was to determine the genotypes/subgenotypes of the hepatitis B virus and establish their role in maintaining the epidemic process in the country. Serological (CMIA, ELISA), molecular biological (PCR, sequencing) and bioinformatic (phylogenetic analysis) methods have been used to obtain results. As studies have shown, in 722 (81.7%) samples genotype D has been determined; in 156 (17.7%) - genotype A; in 3 (0.3%) - genotype C and only in 1 sample (0.1%) – genotype B. In two cases (0.2%) a recombinant form of the hepatitis B virus has been detected. The epidemic process of hepatitis B in the country is supported mainly by the circulation of “local” variants of the virus. At the same time, there are occasional introductions of new genetic variants from the outside of the country.
Keywords. HBV, genotypes/subgenotypes, sequencing, phylogenetic analysis, hepatitis B.
Introduction. Hepatitis B virus (HBV) infection remains a serious health concern despite the HBV vaccine introducing in the late 1990s. Approximately two billion cases of the disease were reported worldwide from the time of HBV discovery to the beginning of the 21st century. More than 400 million of them subsequently became chronic HBV carriers [1]. Chronic HBV infection is considered a major cause of many liver-related medical complications such as hepatocellular carcinoma, liver cirrhosis, and liver failure [2,3,4]. Due to the virus complexity, HBV genome contains an overlapping region that encodes four different genes (S, C, P and X). The S gene encodes exclusively the HBV surface envelope protein in its long open reading frame [6]. The presence of start codons allows the gene splicing into small, medium and large polypeptides, which are typically used either as a single target gene or in combination with other HBV genes to detect HBV DNA [5]. HBV core antigen is commonly encoded by the C gene, whereas the X gene encodes protein X [7]. The P gene encodes a polymerase protein, an integral component of the reverse transcription process during HBV replication. The HBV replication cycle lacks proofreading properties, resulting in progeny with a high genomic variability [5]. Due to significant genetic diversity, HBV is categorized into ten genotypes (A-J) with 7.5 % intergroup variation [8]. In addition to E and G, all genotypes are classified into 25 subgenotypes with 4% amino acid variability [9,10]. HBV genotypes are distributed differently depending on the geography: HBV-B, HBV-C and HBV-E are most common in Oceania and East Asia, while HBV-E is most prevalent in Central and West Africa. HBV-F and HBV H are found only in Alaska and Latin America. In contrast, HBV-D is a global pandemic. In Australia, Europe, Indonesia, North Africa and Western Asia, HBV-D1 is the most common virus, while HBV-D2 is found in Albania, Japan, Malaysia, North-Eastern Europe, Russia and the UK [11,12,13,14,15]. The progression and natural course of the disease varies among different HBV genotypes. These factors can complicate HBV infection treatment since the efficacy of known therapeutics turns out to be ineffective against certain genotypes and new genotypic variants [8,9]. Thus, there is a global exigency for constant update in genotypic information and surveys of HBV-infected patients [15]. The Republic of Belarus is a country with a moderate level of prevalence of parenteral viral hepatitis B. The inclusion of the vaccine against HBV infection into the Preventive Vaccination Schedule has made it possible to reduce the incidence of acute hepatitis B in the country by 6 times over the past 10 years and allows currently to consider Belarus as a country with a low level of prevalence of acute hepatitis B (less than 2% of the population). The most affected age groups are teenagers and adults 15 to 40 years old. The incidence rate in this specific age group is significantly higher than in other age groups of the population of the Republic. Approximately 800-900 new cases of chronic viral hepatitis B are detected annually in the country, despite a significant decrease of occurrence of acute viral hepatitis B. According to the epidemiological essays, 75,000 patients with chronic hepatitis B currwntly live in Belarus.Materials and methods Samples. 884 blood serum/plasma samples were tested for HBV markers with CMIA /ELISA and PCR methods. The samples were obtained from patients and blood donors with acute and chronic HBV from different regions of the country. The age of the patients ranged from 4 years to 78 years. 404 samples were obtained from males and 250 – from females. In 230 cases gender and age were not specified. Serum/plasma samples were collected and analyzed during 2014-2023. Serological tests. The commercial kit chemiluminescent microparticle immunoassay (CMIA) Architect HBsAg Qualitative II Reagent Kit, Abbott, USA was applied to detect HBsAg. Positive results were confirmed using the Architect HBsAg Qualitative II Confirmatory Reagent Kit, Abbott, USA as well as ELISA “Vectogep B-HBs antigen», manufactured by “Vector-Best”, Novosibirsk, Russia. PCR. The polymerase chain reaction was used to determine DNA of the hepatitis B virus using test systems, produced by Vector-Best CJSC: “RealBest HBV DNA (quantitative”, “RealBest HIV RNA (quantitative”) in accordance with the manufacturer’s instructions. Primers. We used the previously described by L Serfaty et al [16] primer pairs for sequencing the S and P regions of the hepatitis B virus genome. DNA isolation. Viral RNA/DNA from the blood serum/plasma samples were isolated using the “Kit of reagents for the isolation of NK” (manufactured by “Vector-Best”, Russia) and the “Kit of reagents for the isolation of RNA/DNA from clinical material” “RIBO- sorb", "RNA-prep", Russia. The manufacturers’ instructions were strictly followed. Nested PCR was performed using previously described primer pairs p1/pR5 (position 1197-1178) and p4/pR2 (1017-997) [16] according to the following protocol in a volume of 25µl: 2,5 µl 10х buffer+ MgCl2; 0,25 µl 25mM dNTPs; 0,5 µl 10 µM p1; 0,5 µl 10 µM pR5; 0,8 10 µM Taq polymerase; 18,45 µl bdH20. The first round of PCR was carried out according to the following protocol: one denaturation cycle at 95oC for 2 minutes and 30 cycles of amplification were performed with denaturation at 95°C for 30 sec, annealing at 53°C for 30 sec, and extension at 68°C for 1 minutes, then one cycle at 68oC for 5 minutes and storage at 10oC. In the second round of PCR, the annealing temperature was 50°C; the rest parameters were not changed. Amplified DNA fragments were analyzed on a 2% agarose gel. Electrophoresis was carried out at 10 w/cm of gel in TRIS-acetate buffer, pH 8.0. DNA was visualized using the Vitran Photo gel documentation system (Biocom Company LLC, Russia). The fragment size was determined according to the molecular weight of marker of 100-1000 bp (Fermentas, Lithuania).
The resulting DNA fragments were purified using the NimaGen ExS-Pure kit, Holland.
PCR sequencing was carried out using the second round primer pair p4/pR2 in the volume of 10 µl according to the following protocol: 5x seqbuf - 2 µl; BigDye Terminator v. 3.1 (applied biosystems) - 1 µl; p4/pR4 - 2 µl; bdH20 - 3 bdH20; DNA - 2 µl.
Electrophoresis of HBV DNA fragments obtained and purified after PCR sequencing was performed on AB 3500 genetic analyzer, USA. Phylogenetic analysis of the obtained HBV DNA fragments was conducted using the computer programs: Sequencing Analysis v.6, BioEdit, SeqScape v3, MEGAX and Genious 8.1. Phylogenetic trees were built using the ML (maximum likelihood) algorithm in the PHYML program. The SH-aLRT test was performed to calculate the statistical significance of the clusters. Clusters with a support node ≥0.9 were considered reliable. Mutations in the HBV genome in the S and P regions were determined using the programs: https://www.geno2pheno.org, http://www.hiv-grade.de/hbv_grade and https://hivdb.stanford.edu/HBV/HBVseq. Results The studies showed the following: 722 (81.7%) out of all 884 sequenced and analyzed samples, had genotype D; 156 (17.7%) - genotype A; 3 (0.3%) - genotype C and the only 1 (0.1%) – genotype B. For two cases (0.2%) a recombinant form of the hepatitis B virus was detected. Genotype D was found to be represented by subgenotypes - D1 (145/20.1%), D2 (371/51.4%), D3 (201/27, 8%) and D4 (5/0.7%); A – A2; C – C1 and C2; B – B4; recombinant forms C2/ D and A2/ С2 were detected (Figure 1). Figure 1. HBV subgenotypes identified in Belarus In 12 cases, the virus with mutation mutations to reverse transcriptase inhibitors Lamivudine, Zeffix® and Telbivudine, Tyzeka®, Sebivo® and partial resistance to Entecavir, Baraclude® was identified. Moreover, in 4 cases, resistance of mutations was identified in subgenotypes A2 and D2, in 3 cases- in the genome of subgenotype D1, and in one case - in subgenotype D3. Mutations were most often recorded at positions 180M and 204V - in 8 cases, 204I - in three cases, 80I and 173L - 2 times each, and 80V and 181T in one case (Table 1). Table 1 – Resistance of mutations identified in different HBV subgenotypes
|
N |
Subgenotypes |
Mutations |
|
1. |
А2 |
80V-1; 180M-3; 181T-1; 204V-3; |
|
2. |
D1 |
173L-1; 180M-3; 204V-3; |
|
3. |
D2 |
80I-2; 173L-1; 180M-2; 204V-2; 204I-2; |
|
4. |
D3 |
204I-1; |
Determining mutations is very important for infectious disease doctors, since substitutions at positions 180M, 181T, 204V and 204I lead to resistance to Lamivudine (Zeffix), Adefovir (Hepsera) and Telbivudine (Tyzeka, Sebivo) and require a change in treatment regimen.
Phylogenetic analysis of subtype A2 DNA sequences showed that most samples from Belarus were found nearby and formed at least 7 clusters (Figures 2, 3, 4). At the same time, some samples were in different groups and formed clusters with samples from the USA, Cuba, Brazil, similarly to Western European countries, mainly from Poland, Italy, France, Belgium, Germany and some others (Figures 2,3,4).
|
2 |
|
3 |
|
4 |
Fig.2, 3, 4 Phylogenetic analysis of subtype А2 (Here and in other figures, samples from Belarus are indicated in red)
12 clusters were identified while analyzing samples of subtype D1. Most of the sequenced and analyzed HBV DNA samples were identified with sequences from Pakistan, India, Iran, Bangladesh and some European countries such as France, Holland, Italy. Two large clusters, consisting mainly of samples from Belarus, were found with sequences from Iran and Italy, as well as Pakistan and France, Oman, Holland and Russia (Figures 5,6).
|
5 |
|
6 |
Fig. 5 and 6 Phylogenetic analysis of subtype D1
Sequences of subtype D2 formed 14 clusters, which were present in the samples from Belarus described earlier. They showed similarities with samples from Estonia, USA, India, Spain, Russia, Cameroon, Latvia, Iran and Belgium, Figure 7, 8, 9.
|
7 |
|
8 |
|
9 |
Fig. 7, 8, 9 Phylogenetic analysis of subtype D2
Phylogenetic analysis of D3 subtype showed that the samples formed 14 clusters mainly originated from India, Croatia, Russia, Italy, Germany and even Brazil. The largest cluster included samples from Belarus, with the previously described sequences. These clusters originated from Poland, Russia, Rwanda, Estonia and Albania, Figure 10.
Fig. 10 Phylogenetic analysis of subtype D3
The phylogenetic analysis of the subtype D4 samples showed that they all were found in the same group and clustered with HBV sequences from Cuba and Haiti, Figure 11.
Fig. 11 Phylogenetic analysis of subtype D4
Samples of subgenotypes C1 and C2 were identified in adult patients and a child from China. Sequences of subtype C1 formed a cluster, with samples from the USA and Vietnam, and subtype C2 with reference samples from South Korea, USA and China, Figure 12,13.
Fig .12, 13. Phylogenetic analysis of subtypes C1 and C2
The subtype B4 sample was revealed in a student from Vietnam who came to study in our country.
Finally, both samples with recombinant forms of HBV were identified in residents of Belarus
Discussion
Approximately 800-900 new cases of chronic forms of hepatitis B are reported in Belarus annually. Compulsory vaccination, introduced to the country in the mid-90s, produces a certain positive effect on reducing the number of HBV cases. However, around 75,000 patients with chronic hepatitis B are estimated to live in the country at the moment. Our previous study showed that subgenotype D2 (55.8%) was dominant in Belarus, followed by D3 (18,3%), D1 (11,6%) and A2 (11,6%) [17]. The conducted studies show that the structure of genotypes/subgenotypes in the country has actually remained unchanged. Genotype D still dominates in the country and accounts for 81.7% of all analyzed cases. Genotype D was found to be represented by subgenotypes D1 (145/20.1%), D2 (371/51.4%), D3 (201/27, 8%) and D4 (5/0.7%); A – A2; C – C2; B – B4; recombinant forms C2/ D and A2/ С2 were detected. Possible causes of the infection in patients were determined i.e. they occurred due to the circulation of “domestic” variants of the virus and/or the introduction of the virus from outside. The findings obtained show that the epidemic transmission of viral hepatitis B in the country is maintained mainly due to the circulation of local variants of the virus. It is confirmed by the examples of subgenotypes A2, D1, D2 and D3, where the largest clusters were formed by samples from Belarus. However, it should be noted that new variants of HBV are constantly being introduced into the country. The main routes of introduction of A2 subgenotype are Western Europe countries and even the USA, Cuba, Brazil; for D1 - Pakistan, India, Iran, Bangladesh, Italy, Holland and Russia; D2 - Estonia, USA, India, Spain, Russia, Cameroon, Latvia, Iran and Belgium; D3 - Poland, Russia, Rwanda, Estonia and Albania. All samples of D4 subgenotype were clustered with sequences from Cuba and Haiti, C1 and C2 – from China, and B4 – from Vietnam. Two recombinant forms C2/D and A2/D were identified in the residents of our country. Comparing the distribution of different HBV genotypes in neighboring countries and even in different regions of the same country, we may conclude that they vary. For example, genotype D significantly dominates in the European part of Russian Federation, while, for example, in Yakutia, genotype A occupies one of the leading positions (36.4% - A and 58,6% - D) [18]. In Asia, particularly in its eastern part, genotypes B and C dominate, and the C2 subgenotype is epidemic in China [19, 20]. In Western Europe, in particular in Portugal HBV genotype A (HBV/A) was the most prevalent genotype (41.5%), followed by D [HBV/D; (33.8%)], and E [HBV/E; (24.6%)]. Subgenotypes A1 (HBV/A1) and D4 (HBV/D4) were almost equally prevalent with 23.1% and 22.3%, respectively, followed by A2 (HBV/A2) with 16.2% and D3 (HBV/D3) with 11.5% [21].
The Republic of Belarus is geographically located in the center of Europe. The residents of the country are actively travelling and many people from other countries relocate to Belarus for work and study. This fact can easily explain the introduction of the new variants of HBV to the Republic.
The results of this study contribute valuable information of the molecular genetic characteristics of HBV, prevailing in the Republic of Belarus. Possible routes of the introduction of the new variants of HBV have been identified. The main mutations in the P region of the virus genome leading to resistance to antiviral drugs have detected. Identification of the mutations at positions 181T, 204V, 204I, which determine resistance not only to Lamivudine (Zefffix), but also to Adefovir (Hepsera) and Telbivudine (Tyzeka, Sebivo), made it possible to use Tenofovir DF for treatment of the patients. The compensatory mutations 80V, 173L, 80I have been found in the complex with the mutations leading to virus resistance to antiviral drugs.
Conclusions
To summarize, should be noted that the use of the methods of molecular epidemiology solve many issues for both: epidemiologists and infectious diseases specialists. This knowledge is essential to plan preventive measures for the directions and routes of its’ spread in the country by determining the genotypes/subgenotypes of the hepatitis B virus. The quality of living of our patients could be improved by prescribing new treatment regimens, based on the resistance mutations identification.
REFERENCES
- Thun M.J., DeLancey J.O., Center M.M., Jemal A., Ward E.M. The global burden of cancer: Priorities for prevention. Carcinogenesis. 2010;31:100–110.
- Philips C.A., Ahamed R., Abduljaleel J.K., Rajesh S., Augustine P. Critical Updates on Chronic Hepatitis B Virus Infection in 2021. Cureus. 2021;13:e19152.
- Kao J.-H., Chen P.-J., Lai M.-Y., Chen D.-S. Hepatitis B genotypes correlate with clinical outcomes in patients with chronic hepatitis B. Gastroenterology. 2000;118:554–559.
- Aghakhani A., Hamkar R., Zamani N., Eslamifar A., Banifazl M., Saadat A., Sofian M., Adibi L., Irani N., Mehryar M. Hepatitis B virus genotype in Iranian patients with hepatocellular carcinoma. Int. J. Infect. Dis. 2009;13:685–689.
- Beck J., Nassal M. Hepatitis B virus replication. World J. Gastroenterol. 2007;13:48–64.
- Pollicino T., Cacciola I., Saffioti F., Raimondo G. Hepatitis B virus PreS/S gene variants: Pathobiology and clinical implications. J. Hepatol. 2014;61:408–417.
- Ie S.I., Thedja M.D., Roni M., Muljono D.H. Prediction of conformational changes by single mutation in the hepatitis B virus surface antigen (HBsAg) identified in HBsAg-negative blood donors. Virol. J. 2010;7:326.
- Lin C.-L., Kao J.-H. The clinical implications of hepatitis B virus genotype: Recent advances. J. Gastroenterol. Hepatol. 2011;26((Suppl. 1)):123–130.
- Zhu S.S., Zhang H.F., Wang H.B., Dong Y., Chen D., Jia W., Gan Y., Chen J. Relation of viral genotypes to clinical features in children with chronic hepatitis B. Chin. J. Exp. Clin. Virol. 2008;22:192–194.
- Shamseer L., Moher D., Clarke M., Ghersi D., Liberati A., Petticrew M., Shekelle P., Stewart L.A. Preferred reporting items for systematic review and meta-analysis protocols (PRISMA-P) 2015: Elaboration and explanation. BMJ. 2015;349:g7647.
- George B.J., Aban I.B. An application of meta-analysis based on DerSimonian and Laird method. J. Nucl. Cardiol. 2016;23:690–692.
- Fletcher J. What is heterogeneity and is it important? BMJ. 2007;334:94–96.
- Higgins J.P.T., Thompson S.G., Deeks J.J., Altman D.G. Measuring inconsistency in meta-analyses. BMJ. 2003;327:557–560.
- Wallace B.C., Dahabreh I.J., Trikalinos T.A., Lau J., Trow P., Schmid C.H. Closing the Gap between Methodologists and End-Users: R as a Computational Back-End. J. Stat. Softw. 2012;49:2–15.
- Munn Z., MClinSc S.M., Lisy K., Riitano D., Tufanaru C. Methodological guidance for systematic reviews of observational epidemiological studies reporting prevalence and cumulative incidence data. Int. J. Evid. Based Healthc. 2015;13:147–153.
- Serfaty L., Thabut D., Zoulim F., Andreani T., Eres O. C., Carbonell N., Loria A., and Poupon R. / Hepatology – 2001 – V.34 – N.- 3 – P.573-577.
- Olinger C.M., Lazouskaya N.V., Eremin V.F., Muller C.P. Multiple genotypes of hepatitis B and C viruses in Belarus: similarities with Russia and European influences. Clin Microbiol Infect – 2008; 14:575-581.
- Anastasia A Karlsen, Karen K Kyuregyan, Olga V Isaeva, Vera S Kichatova2, Fedor A Asadi Mobarkhan, Lyudmila V Bezuglova, Irina G Netesova, Victor A Manuylov, Andrey A Pochtovyi, Vladimir A Gushchin, Snezhana S Sleptsova, Margarita E Ignateva, Mikhail I Mikhailov Different evolutionary dynamics of hepatitis B virus genotypes A and D, and hepatitis D virus genotypes 1 and 2 in an endemic area of Yakutia, Russia // /BMC Infect Dis – 2022; 12;22:452
- Kizito Eneye Bello, Tuan Nur Akmalina Mat Jusoh, Ahmad Adebayo Irekeola, Norhidayah Abu, Nur Amalin Zahirah Mohd Amin, Nazri Mustaffa, Rafidah Hanim Shueb/ A Recent Prevalence of Hepatitis B Virus (HBV) Genotypes and Subtypes in Asia: A Systematic Review and Meta-Analysis // Healthcare (Basel) – 2023; 1;11(7):1011.
- Bing Sun, Aida Andrades Valtueña, Arthur Kocher, Shizhu Gao, Chunxiang Li, Shuang Fu, Fan Zhang, Pengcheng Ma, Xuan Yang, Yulan Qiu, Quanchao Zhang, Jian Ma, Shan Chen, Xiaoming Xiao, Sodnomjamts Damchaabadgar, Fajun Li, Alexey Kovalev, Chunbai Hu, Xianglong Chen, Lixin Wang, Wenying Li, Yawei Zhou, Hong Zhu Johannes Krause, Alexander Herbig, Yinqiu Cui/ Origin and dispersal history of Hepatitis B virus in Eastern Eurasia // Nat Commun, 2024; 5;15:2951.
- Rute Marcelino, Ifeanyi Jude Ezeonwumelu, André Janeiro, Paula Mimoso, Sónia Matos, Veronica Briz, Victor Pimentel, Marta Pingarilho, Rui Tato Marinho, José Maria Marcelino, Nuno Taveira, Ana Abecasis / . Phylogeography of hepatitis B virus: The role of Portugal in the early dissemination of HBVworldwide // PLoS One. 2022; 17(12):
Acknowledgments
We would like to show our gratitude to the Abbott Transfusion Medicine Medical affairs team for financial support of the publication.
Abstract. Annually 800-900 new cases of hepatitis B chronic forms are registered in Belarus. Compulsory vaccination introduced in the mid-90s certainly produces a positive effect on reducing the number of HBV cases. About 75,000 patients with chronic hepatitis B living in the country are estimated to live in the country at the moment. The main goal of this work was to determine the genotypes/subgenotypes of the hepatitis B virus and establish their role in maintaining the epidemiological process on the territory of Belarus. Serological (CMIA, ELISA), molecular biological (PCR, sequencing) and bioinformatic (phylogenetic analysis) methods have been used. As studies have shown, out of all 884 sequenced and analyzed samples, 722 (81.7%) of them have genotype D; 156 (17.7%) - genotype A; 3 (0.3%) - genotype C and the only 1 (0.1%) – genotype B. For two cases (0.2%) a recombinant form of the hepatitis B virus was detected. The epidemic process of hepatitis B in the country is supported mainly by the circulation of “local” variants of the virus. At the same time, there are occasional introductions of new variants from outside the country.
Keywords. HBV, genotypes/subgenotypes, sequencing, phylogenetic analysis.
|
Response to Reviewer 1 Comments
|
||
|
1. Summary |
|
|
|
Thank you very much for taking the time to review this manuscript. Please find the detailed responses below and the corresponding revisions/corrections highlighted/in track changes in the re-submitted files.
|
||
|
2. Questions for General Evaluation |
Reviewer’s Evaluation |
Response and Revisions |
|
Does the introduction provide sufficient background and include all relevant references? |
Yes/Can be improved/Must be improved/Not applicable |
|
|
Are all the cited references relevant to the research? |
Yes/Can be improved/Must be improved/Not applicable |
|
|
Is the research design appropriate? |
Yes/Can be improved/Must be improved/Not applicable |
|
|
Are the methods adequately described? |
Yes/Can be improved/Must be improved/Not applicable |
|
|
Are the results clearly presented? |
Yes/Can be improved/Must be improved/Not applicable |
|
|
Are the conclusions supported by the results? |
Yes/Can be improved/Must be improved/Not applicable |
|
|
3. Point-by-point response to Comments and Suggestions for Authors |
||
|
Comments 1: The Abstract section is incomplete, which only contains results. Please include a few sentences of introduction, methods and materials, and conclusion.
|
||
|
Response 1: [Type your response here and mark your revisions in red] Thank you for pointing this out. I/We agree with this comment. Therefore, I/we have supplemented the introduction, materials and methods, and conclusion. Abstract. Annually 800-900 new cases of hepatitis B chronic forms are registered in Belarus. Compulsory vaccination introduced in the mid-90s certainly produces a positive effect on reducing the number of HBV cases. About 75,000 patients with chronic hepatitis B living in the country are estimated to live in the country at the moment. The main goal of this work was to determine the genotypes/subgenotypes of the hepatitis B virus and establish their role in maintaining the epidemiological process on the territory of Belarus. Serological (CMIA, ELISA), molecular biological (PCR, sequencing) and bioinformatic (phylogenetic analysis) methods have been used. As studies have shown, out of all 884 sequenced and analyzed samples, 722 (81.7%) of them have genotype D; 156 (17.7%) - genotype A; 3 (0.3%) - genotype C and the only 1 (0.1%) – genotype B. For two cases (0.2%) a recombinant form of the hepatitis B virus was detected. The epidemic process of hepatitis B in the country is supported mainly by the circulation of “local” variants of the virus. At the same time, there are occasional introductions of new variants from outside the country.
|
||
|
Comments 2: Introduction section: Background information of HBV infection in Belarus is needed.
|
||
|
Response 2: Agree. I/We have, accordingly, done/revised/changed/modified…..to emphasize this point. In the introduction, I inserted information on HBV in Belarus. Also in the discussion there is some data on the spread of HBV in our country Comments 3. Materials and Methods section: Consider dividing the whole section into several parts with subtitles for a better readability. Response 3. In the materials and methods section, I tried to highlight sections for better understanding. Comments 4. Results section: Figures 2 to 12 are hard to comprehend. Please specify the red labels in each figure legend. Additionally, the letters are too small to read. Response 4. I agree that the figures are small and I marked all Belarusian samples in red and signed under figures 2,3 and 4 that these are samples from Belarus. I have highlighted certain fragments for a better understanding of the description. If you leave the trees entirely, then for D2 the drawing takes up three sheets.Comments 5. In the phylogenetic trees, the numbers in the branches are unnecessary to be shown. Some representative ones should be enough. Response 5. I left the numbers to assess statistical significance and assess the degree of relatedness of the samples in the cluster. I will describe this in detail in the next article. Comments 6. The authors should upload all relevant HBV sequences derived from this study to the GenBank database. Response 6. We are currently submitting our sequences to the GenBank database. Comments 7. Discussion section: I wonder why there is no discussion of HBV mutations since you performed mutational analysis.
Response 7. In the discussion section I inserted material on mutations Comments 8. Please standardize the abbreviation of hepatitis B virus as HBV throughout the manuscript. Response 8. I left the designation HBV in the text
|
||
|
4. Response to Comments on the Quality of English Language |
||
|
Point 1: |
||
|
Response 1: We have improved the English text and hope it is more readable.
|
||
|
5. Additional clarifications |
||
|
[Here, mention any other clarifications you would like to provide to the journal editor/reviewer.] |
||
Reviewer 2 Report
Comments and Suggestions for Authors
Abstract: Clearly define abbreviations such as HBV and PCR upon first use to improve readability.
Introduction: Add context on the importance of studying HBV genotypes specifically in Belarus and compare this with neighbouring regions to emphasize the study’s relevance.
Methods:
· Sample Selection: Clarify criteria for sample inclusion, such as a minimum viral load. Mention how missing demographic data (e.g., age/gender for 230 samples) were addressed to ensure data reliability.
· Specify if the samples were randomly selected or stratified by demographics, which would clarify representativeness.
· Describe contamination control measures used during PCR to enhance confidence in sample integrity.
· Line 90: Add a brief note on the specificity of the primers, indicating whether they were designed for Belarus-specific strains or universally applicable strains.
· Phylogenetic Analysis: Briefly explain the rationale for using the maximum likelihood (ML) method over other approaches and include software settings to improve reproducibility.
Genotype Distribution Results:
· Add confidence intervals for genotype frequencies to improve precision.
· Provide a brief comparison of genotype distribution in Belarus with trends observed in nearby regions.
Discussion:
· Discuss the implications of genotype diversity on treatment, especially regarding drug resistance.
· Mention potential clinical outcomes of mutations identified, including any resistance levels to standard treatments.
· Highlight potential impacts on public health policies in Belarus, such as vaccination and monitoring strategies.
Limitations: Acknowledge limitations, such as possible sampling bias or geographic constraints, to provide a balanced view of the findings.
Additional Minor Comments:
Line 42: Briefly mention any potential sampling bias due to missing demographic data, which may affect the generalizability of findings.
Table 1 (Mutation Data): Consider bolding mutations linked to drug resistance to make key findings more noticeable.
Figures (Phylogenetic Trees): Label key clusters in figures, especially those with notable regional associations, for easier interpretation by readers unfamiliar with Belarusian data.
Data Sources: Reference primary data sources (e.g., national health databases) to ensure transparency in data collection.
Comments on the Quality of English LanguageRevising the manuscript’s language and correcting grammatical errors will improve clarity, professionalism, and readability. making the research more accessible and compelling to readers.
Author Response
|
Response to Reviewer 2 Comments
|
|||||||||||||||||||||||||
|
1. Summary |
|
|
|||||||||||||||||||||||
|
Thank you very much for taking the time to review this manuscript. Please find the detailed responses below and the corresponding revisions/corrections highlighted/in track changes in the re-submitted files. |
|||||||||||||||||||||||||
|
2. Questions for General Evaluation |
Reviewer’s Evaluation |
Response and Revisions |
|||||||||||||||||||||||
|
Does the introduction provide sufficient background and include all relevant references? |
Yes/Can be improved/Must be improved/Not applicable |
|
|||||||||||||||||||||||
|
Are all the cited references relevant to the research? |
Yes/Can be improved/Must be improved/Not applicable |
|
|||||||||||||||||||||||
|
Is the research design appropriate? |
Yes/Can be improved/Must be improved/Not applicable |
|
|||||||||||||||||||||||
|
Are the methods adequately described? |
Yes/Can be improved/Must be improved/Not applicable |
|
|||||||||||||||||||||||
|
Are the results clearly presented? |
Yes/Can be improved/Must be improved/Not applicable |
|
|||||||||||||||||||||||
|
Are the conclusions supported by the results? |
Yes/Can be improved/Must be improved/Not applicable |
|
|||||||||||||||||||||||
|
3. Point-by-point response to Comments and Suggestions for Authors |
|||||||||||||||||||||||||
| Comments 1: Abstract: Clearly define abbreviations such as HBV and PCR upon first use to improve readability.Response 1: Thank you for pointing this out. I/We agree with this comment. Therefore, I have made a transcription of the first mentioned abbreviations in the text
Comments 2: Introduction: Add context on the importance of studying HBV genotypes specifically in Belarus and compare this with neighbouring regions to emphasize the study’s relevance. Response 2: I modified the introduction and outlined the importance of hepatitis B virus research for Belarus. In the discussion section, I provide data on genotypes previously studied in the country in comparison with neighboring countries. Methods: Comments 3: Sample Selection: Clarify criteria for sample inclusion, such as a minimum viral load. Mention how missing demographic data (e.g., age/gender for 230 samples) were addressed to ensure data reliability. · Specify if the samples were randomly selected or stratified by demographics, which would clarify representativeness. Response 3: The determined viral load level in the sample ranged from 1.0x102 copiesDNA/mL to >8.2x108 copiesDNA/mL (the limit of sensitivity of the test system)604 serum/plasma samples were from Minsk and Minsk region, 71 from Gomel region, 86 from Vitebsk region, 64 from Mogilev region, 51 from Brest region and 8 from Grodno region. Comments 4: Describe contamination control measures used during PCR to enhance confidence in sample integrity. Response 4: There is no need to describe the contamination control, as each test system contains internal controls to track the reaction from DNA extraction to real-time results. When performing sequencing PCR and sequencing, we obtain nucleotide sequences, which are then processed in different programs to build a phylogenetic tree, where we can clearly see whether the viruses (DNA sequences) are different or not. Comments 5: Line 90: Add a brief note on the specificity of the primers, indicating whether they were designed for Belarus-specific strains or universally applicable strains. Response 5: We used the previously described by L Serfaty et al [16] primer pairs for sequencing the S and P regions of the hepatitis B virus genome.Phylogenetic Analysis: Briefly explain the rationale for using the maximum likelihood (ML) method over other approaches and include software settings to improve reproducibility. The ML algorithm is optimal for processing a large number of nucleotide sequences and constructing a phylogenetic tree. We also used the neighbor joining method, but the ML algorithm is more representative. Genotype Distribution Results: Comments 6: Add confidence intervals for genotype frequencies to improve precision. Provide a brief comparison of genotype distribution in Belarus with trends observed in nearby regions. Response 6: In the discussion section, we compare our genotypes with those identified in some neighbouring countries. Discussion: Comments 7: · Discuss the implications of genotype diversity on treatment, especially regarding drug resistance. · Mention potential clinical outcomes of mutations identified, including any resistance levels to standard treatments. · Highlight potential impacts on public health policies in Belarus, such as vaccination and monitoring strategies. Response 7: In the discussion section, I described the importance of mutations in the treatment of patients with hepatitis B. More details about mutations and treatment will be described in the article on treating our patients.Comments 8: Limitations: Acknowledge limitations, such as possible sampling bias or geographic constraints, to provide a balanced view of the findings. Response 8: We characterise genotypes/subgenotypes on the territory of the country with a population of 9.2 million. And a larger number of such patients is concentrated in Minsk and Minsk region. Therefore, it is generally possible to extrapolate the results obtained to the whole country, taking into account that samples were also taken from other regions.Comments 9: Additional Minor Comments: Line 42: Briefly mention any potential sampling bias due to missing demographic data, which may affect the generalizability of findings. Table 1 (Mutation Data): Consider bolding mutations linked to drug resistance to make key findings more noticeable. Figures (Phylogenetic Trees): Label key clusters in figures, especially those with notable regional associations, for easier interpretation by readers unfamiliar with Belarusian data. Data Sources: Reference primary data sources (e.g., national health databases) to ensure transparency in data collection. Response 9: The samples from Belarus and, accordingly, the clusters they form with the sequences taken from GenBank are highlighted in red on the phylagenetic trees. Unfortunately, we cannot provide complete phylogenetic trees because they take many sheets.
|
|||||||||||||||||||||||||
|
|
|||||||||||||||||||||||||
|
|
|||||||||||||||||||||||||
|
|
|||||||||||||||||||||||||
|
4. Response to Comments on the Quality of English Language |
|||||||||||||||||||||||||
|
Point 1: |
|||||||||||||||||||||||||
|
Response 1: We have improved the English text and hope it is more readable.
|
|||||||||||||||||||||||||
|
5. Additional clarifications |
|||||||||||||||||||||||||
|
[Here, mention any other clarifications you would like to provide to the journal editor/reviewer.]
HEPATITIS B VIRUS GENOTYPES AND SUBGENOTYPES CIRCULATING IN BELARUS 1Vladimir Eremin, 1Fedor Karpenko, 2Igor Karpov, 3Valeriy Semenov, 3Ina Oiestad
1Republican Scientific and Practical Center for Transfusiology and Medical Biotechnology 2Belarusian State Medical University, Department of Infectious Diseases, Belarus 3Vitebsk State Medical University, Department of Infectious Diseases, Belarus Abstract. Approximately 800-900 new cases of chronic forms of hepatitis B are reported in Belarus annually. Compulsory vaccination, introduced to the country in the mid-90s, produces a certain positive effect on reducing the number of HBV cases. However, around 75,000 patients with chronic hepatitis B are estimated to live in the country at the moment. The main goal of this research was to determine the genotypes/subgenotypes of the hepatitis B virus and establish their role in maintaining the epidemic process in the country. Serological (CMIA, ELISA), molecular biological (PCR, sequencing) and bioinformatic (phylogenetic analysis) methods have been used to obtain results. As studies have shown, in 722 (81.7%) samples genotype D has been determined; in 156 (17.7%) - genotype A; in 3 (0.3%) - genotype C and only in 1 sample (0.1%) – genotype B. In two cases (0.2%) a recombinant form of the hepatitis B virus has been detected. The epidemic process of hepatitis B in the country is supported mainly by the circulation of “local” variants of the virus. At the same time, there are occasional introductions of new genetic variants from the outside of the country. Keywords. HBV, genotypes/subgenotypes, sequencing, phylogenetic analysis, hepatitis B. Introduction. Hepatitis B virus (HBV) infection remains a serious health concern despite the HBV vaccine introducing in the late 1990s. Approximately two billion cases of the disease were reported worldwide from the time of HBV discovery to the beginning of the 21st century. More than 400 million of them subsequently became chronic HBV carriers [1]. Chronic HBV infection is considered a major cause of many liver-related medical complications such as hepatocellular carcinoma, liver cirrhosis, and liver failure [2,3,4]. Due to the virus complexity, HBV genome contains an overlapping region that encodes four different genes (S, C, P and X). The S gene encodes exclusively the HBV surface envelope protein in its long open reading frame [6]. The presence of start codons allows the gene splicing into small, medium and large polypeptides, which are typically used either as a single target gene or in combination with other HBV genes to detect HBV DNA [5]. HBV core antigen is commonly encoded by the C gene, whereas the X gene encodes protein X [7]. The P gene encodes a polymerase protein, an integral component of the reverse transcription process during HBV replication. The HBV replication cycle lacks proofreading properties, resulting in progeny with a high genomic variability [5]. Due to significant genetic diversity, HBV is categorized into ten genotypes (A-J) with 7.5 % intergroup variation [8]. In addition to E and G, all genotypes are classified into 25 subgenotypes with 4% amino acid variability [9,10]. HBV genotypes are distributed differently depending on the geography: HBV-B, HBV-C and HBV-E are most common in Oceania and East Asia, while HBV-E is most prevalent in Central and West Africa. HBV-F and HBV H are found only in Alaska and Latin America. In contrast, HBV-D is a global pandemic. In Australia, Europe, Indonesia, North Africa and Western Asia, HBV-D1 is the most common virus, while HBV-D2 is found in Albania, Japan, Malaysia, North-Eastern Europe, Russia and the UK [11,12,13,14,15]. The progression and natural course of the disease varies among different HBV genotypes. These factors can complicate HBV infection treatment since the efficacy of known therapeutics turns out to be ineffective against certain genotypes and new genotypic variants [8,9]. Thus, there is a global exigency for constant update in genotypic information and surveys of HBV-infected patients [15]. The Republic of Belarus is a country with a moderate level of prevalence of parenteral viral hepatitis B. The inclusion of the vaccine against HBV infection into the Preventive Vaccination Schedule has made it possible to reduce the incidence of acute hepatitis B in the country by 6 times over the past 10 years and allows currently to consider Belarus as a country with a low level of prevalence of acute hepatitis B (less than 2% of the population). The most affected age groups are teenagers and adults 15 to 40 years old. The incidence rate in this specific age group is significantly higher than in other age groups of the population of the Republic. Approximately 800-900 new cases of chronic viral hepatitis B are detected annually in the country, despite a significant decrease of occurrence of acute viral hepatitis B. According to the epidemiological essays, 75,000 patients with chronic hepatitis B currwntly live in Belarus.Materials and methods Samples. 884 blood serum/plasma samples were tested for HBV markers with CMIA /ELISA and PCR methods. The samples were obtained from patients and blood donors with acute and chronic HBV from different regions of the country. The age of the patients ranged from 4 years to 78 years. 404 samples were obtained from males and 250 – from females. In 230 cases gender and age were not specified. Serum/plasma samples were collected and analyzed during 2014-2023. Serological tests. The commercial kit chemiluminescent microparticle immunoassay (CMIA) Architect HBsAg Qualitative II Reagent Kit, Abbott, USA was applied to detect HBsAg. Positive results were confirmed using the Architect HBsAg Qualitative II Confirmatory Reagent Kit, Abbott, USA as well as ELISA “Vectogep B-HBs antigen», manufactured by “Vector-Best”, Novosibirsk, Russia. PCR. The polymerase chain reaction was used to determine DNA of the hepatitis B virus using test systems, produced by Vector-Best CJSC: “RealBest HBV DNA (quantitative”, “RealBest HIV RNA (quantitative”) in accordance with the manufacturer’s instructions. Primers. We used the previously described by L Serfaty et al [16] primer pairs for sequencing the S and P regions of the hepatitis B virus genome. DNA isolation. Viral RNA/DNA from the blood serum/plasma samples were isolated using the “Kit of reagents for the isolation of NK” (manufactured by “Vector-Best”, Russia) and the “Kit of reagents for the isolation of RNA/DNA from clinical material” “RIBO- sorb", "RNA-prep", Russia. The manufacturers’ instructions were strictly followed. Nested PCR was performed using previously described primer pairs p1/pR5 (position 1197-1178) and p4/pR2 (1017-997) [16] according to the following protocol in a volume of 25µl: 2,5 µl 10х buffer+ MgCl2; 0,25 µl 25mM dNTPs; 0,5 µl 10 µM p1; 0,5 µl 10 µM pR5; 0,8 10 µM Taq polymerase; 18,45 µl bdH20. The first round of PCR was carried out according to the following protocol: one denaturation cycle at 95oC for 2 minutes and 30 cycles of amplification were performed with denaturation at 95°C for 30 sec, annealing at 53°C for 30 sec, and extension at 68°C for 1 minutes, then one cycle at 68oC for 5 minutes and storage at 10oC. In the second round of PCR, the annealing temperature was 50°C; the rest parameters were not changed. Amplified DNA fragments were analyzed on a 2% agarose gel. Electrophoresis was carried out at 10 w/cm of gel in TRIS-acetate buffer, pH 8.0. DNA was visualized using the Vitran Photo gel documentation system (Biocom Company LLC, Russia). The fragment size was determined according to the molecular weight of marker of 100-1000 bp (Fermentas, Lithuania). The resulting DNA fragments were purified using the NimaGen ExS-Pure kit, Holland. PCR sequencing was carried out using the second round primer pair p4/pR2 in the volume of 10 µl according to the following protocol: 5x seqbuf - 2 µl; BigDye Terminator v. 3.1 (applied biosystems) - 1 µl; p4/pR4 - 2 µl; bdH20 - 3 bdH20; DNA - 2 µl. Electrophoresis of HBV DNA fragments obtained and purified after PCR sequencing was performed on AB 3500 genetic analyzer, USA. Phylogenetic analysis of the obtained HBV DNA fragments was conducted using the computer programs: Sequencing Analysis v.6, BioEdit, SeqScape v3, MEGAX and Genious 8.1. Phylogenetic trees were built using the ML (maximum likelihood) algorithm in the PHYML program. The SH-aLRT test was performed to calculate the statistical significance of the clusters. Clusters with a support node ≥0.9 were considered reliable. Mutations in the HBV genome in the S and P regions were determined using the programs: https://www.geno2pheno.org, http://www.hiv-grade.de/hbv_grade and https://hivdb.stanford.edu/HBV/HBVseq. Results The studies showed the following: 722 (81.7%) out of all 884 sequenced and analyzed samples, had genotype D; 156 (17.7%) - genotype A; 3 (0.3%) - genotype C and the only 1 (0.1%) – genotype B. For two cases (0.2%) a recombinant form of the hepatitis B virus was detected. Genotype D was found to be represented by subgenotypes - D1 (145/20.1%), D2 (371/51.4%), D3 (201/27, 8%) and D4 (5/0.7%); A – A2; C – C1 and C2; B – B4; recombinant forms C2/ D and A2/ С2 were detected (Figure 1). Figure 1. HBV subgenotypes identified in Belarus In 12 cases, the virus with mutation mutations to reverse transcriptase inhibitors Lamivudine, Zeffix® and Telbivudine, Tyzeka®, Sebivo® and partial resistance to Entecavir, Baraclude® was identified. Moreover, in 4 cases, resistance of mutations was identified in subgenotypes A2 and D2, in 3 cases- in the genome of subgenotype D1, and in one case - in subgenotype D3. Mutations were most often recorded at positions 180M and 204V - in 8 cases, 204I - in three cases, 80I and 173L - 2 times each, and 80V and 181T in one case (Table 1). Table 1 – Resistance of mutations identified in different HBV subgenotypes
Determining mutations is very important for infectious disease doctors, since substitutions at positions 180M, 181T, 204V and 204I lead to resistance to Lamivudine (Zeffix), Adefovir (Hepsera) and Telbivudine (Tyzeka, Sebivo) and require a change in treatment regimen. Phylogenetic analysis of subtype A2 DNA sequences showed that most samples from Belarus were found nearby and formed at least 7 clusters (Figures 2, 3, 4). At the same time, some samples were in different groups and formed clusters with samples from the USA, Cuba, Brazil, similarly to Western European countries, mainly from Poland, Italy, France, Belgium, Germany and some others (Figures 2,3,4).
Fig.2, 3, 4 Phylogenetic analysis of subtype А2 (Here and in other figures, samples from Belarus are indicated in red)
12 clusters were identified while analyzing samples of subtype D1. Most of the sequenced and analyzed HBV DNA samples were identified with sequences from Pakistan, India, Iran, Bangladesh and some European countries such as France, Holland, Italy. Two large clusters, consisting mainly of samples from Belarus, were found with sequences from Iran and Italy, as well as Pakistan and France, Oman, Holland and Russia (Figures 5,6).
Fig. 5 and 6 Phylogenetic analysis of subtype D1 Sequences of subtype D2 formed 14 clusters, which were present in the samples from Belarus described earlier. They showed similarities with samples from Estonia, USA, India, Spain, Russia, Cameroon, Latvia, Iran and Belgium, Figure 7, 8, 9.
Fig. 7, 8, 9 Phylogenetic analysis of subtype D2 Phylogenetic analysis of D3 subtype showed that the samples formed 14 clusters mainly originated from India, Croatia, Russia, Italy, Germany and even Brazil. The largest cluster included samples from Belarus, with the previously described sequences. These clusters originated from Poland, Russia, Rwanda, Estonia and Albania, Figure 10.
Fig. 10 Phylogenetic analysis of subtype D3 The phylogenetic analysis of the subtype D4 samples showed that they all were found in the same group and clustered with HBV sequences from Cuba and Haiti, Figure 11.
Fig. 11 Phylogenetic analysis of subtype D4
Samples of subgenotypes C1 and C2 were identified in adult patients and a child from China. Sequences of subtype C1 formed a cluster, with samples from the USA and Vietnam, and subtype C2 with reference samples from South Korea, USA and China, Figure 12,13.
Fig .12, 13. Phylogenetic analysis of subtypes C1 and C2 The subtype B4 sample was revealed in a student from Vietnam who came to study in our country. Finally, both samples with recombinant forms of HBV were identified in residents of Belarus
Discussion Approximately 800-900 new cases of chronic forms of hepatitis B are reported in Belarus annually. Compulsory vaccination, introduced to the country in the mid-90s, produces a certain positive effect on reducing the number of HBV cases. However, around 75,000 patients with chronic hepatitis B are estimated to live in the country at the moment. Our previous study showed that subgenotype D2 (55.8%) was dominant in Belarus, followed by D3 (18,3%), D1 (11,6%) and A2 (11,6%) [17]. The conducted studies show that the structure of genotypes/subgenotypes in the country has actually remained unchanged. Genotype D still dominates in the country and accounts for 81.7% of all analyzed cases. Genotype D was found to be represented by subgenotypes D1 (145/20.1%), D2 (371/51.4%), D3 (201/27, 8%) and D4 (5/0.7%); A – A2; C – C2; B – B4; recombinant forms C2/ D and A2/ С2 were detected. Possible causes of the infection in patients were determined i.e. they occurred due to the circulation of “domestic” variants of the virus and/or the introduction of the virus from outside. The findings obtained show that the epidemic transmission of viral hepatitis B in the country is maintained mainly due to the circulation of local variants of the virus. It is confirmed by the examples of subgenotypes A2, D1, D2 and D3, where the largest clusters were formed by samples from Belarus. However, it should be noted that new variants of HBV are constantly being introduced into the country. The main routes of introduction of A2 subgenotype are Western Europe countries and even the USA, Cuba, Brazil; for D1 - Pakistan, India, Iran, Bangladesh, Italy, Holland and Russia; D2 - Estonia, USA, India, Spain, Russia, Cameroon, Latvia, Iran and Belgium; D3 - Poland, Russia, Rwanda, Estonia and Albania. All samples of D4 subgenotype were clustered with sequences from Cuba and Haiti, C1 and C2 – from China, and B4 – from Vietnam. Two recombinant forms C2/D and A2/D were identified in the residents of our country. Comparing the distribution of different HBV genotypes in neighboring countries and even in different regions of the same country, we may conclude that they vary. For example, genotype D significantly dominates in the European part of Russian Federation, while, for example, in Yakutia, genotype A occupies one of the leading positions (36.4% - A and 58,6% - D) [18]. In Asia, particularly in its eastern part, genotypes B and C dominate, and the C2 subgenotype is epidemic in China [19, 20]. In Western Europe, in particular in Portugal HBV genotype A (HBV/A) was the most prevalent genotype (41.5%), followed by D [HBV/D; (33.8%)], and E [HBV/E; (24.6%)]. Subgenotypes A1 (HBV/A1) and D4 (HBV/D4) were almost equally prevalent with 23.1% and 22.3%, respectively, followed by A2 (HBV/A2) with 16.2% and D3 (HBV/D3) with 11.5% [21]. The Republic of Belarus is geographically located in the center of Europe. The residents of the country are actively travelling and many people from other countries relocate to Belarus for work and study. This fact can easily explain the introduction of the new variants of HBV to the Republic. The results of this study contribute valuable information of the molecular genetic characteristics of HBV, prevailing in the Republic of Belarus. Possible routes of the introduction of the new variants of HBV have been identified. The main mutations in the P region of the virus genome leading to resistance to antiviral drugs have detected. Identification of the mutations at positions 181T, 204V, 204I, which determine resistance not only to Lamivudine (Zefffix), but also to Adefovir (Hepsera) and Telbivudine (Tyzeka, Sebivo), made it possible to use Tenofovir DF for treatment of the patients. The compensatory mutations 80V, 173L, 80I have been found in the complex with the mutations leading to virus resistance to antiviral drugs.
Conclusions To summarize, should be noted that the use of the methods of molecular epidemiology solve many issues for both: epidemiologists and infectious diseases specialists. This knowledge is essential to plan preventive measures for the directions and routes of its’ spread in the country by determining the genotypes/subgenotypes of the hepatitis B virus. The quality of living of our patients could be improved by prescribing new treatment regimens, based on the resistance mutations identification.
REFERENCES
Acknowledgments We would like to show our gratitude to the Abbott Transfusion Medicine Medical affairs team for financial support of the publication.
Abstract. Annually 800-900 new cases of hepatitis B chronic forms are registered in Belarus. Compulsory vaccination introduced in the mid-90s certainly produces a positive effect on reducing the number of HBV cases. About 75,000 patients with chronic hepatitis B living in the country are estimated to live in the country at the moment. The main goal of this work was to determine the genotypes/subgenotypes of the hepatitis B virus and establish their role in maintaining the epidemiological process on the territory of Belarus. Serological (CMIA, ELISA), molecular biological (PCR, sequencing) and bioinformatic (phylogenetic analysis) methods have been used. As studies have shown, out of all 884 sequenced and analyzed samples, 722 (81.7%) of them have genotype D; 156 (17.7%) - genotype A; 3 (0.3%) - genotype C and the only 1 (0.1%) – genotype B. For two cases (0.2%) a recombinant form of the hepatitis B virus was detected. The epidemic process of hepatitis B in the country is supported mainly by the circulation of “local” variants of the virus. At the same time, there are occasional introductions of new variants from outside the country. Keywords. HBV, genotypes/subgenotypes, sequencing, phylogenetic analysis.
|
|||||||||||||||||||||||||
Reviewer 3 Report
Comments and Suggestions for Authors
Hepatitis B is a vaccine-preventable liver infection caused by the hepatitis B virus. It can cause long-lasting, acute or chronic liver damage. Eremin et al. provide valuable insights for updating the HBV molecular epidemiology in Belarus. However, the manuscript contains many omissions and needs serious revision. The abstract should be rewritten to include an introductory sentence that shows the importance of HBV. The reader can't even orientate himself on what the researched disease is.
Overall, the introduction is well-written.
In the introduction is good to describe the classification of HBV and the genome organization.
At the end of the introduction, it is necessary to include the aim of your study and the period of your research.
Materials and methods are not well described and do not provide complete information about the exodus carried out. Divide your materials and methods into separate points, describing the methods you used coherently and logically. Please distinguish between immunological studies and molecular studies. and provide more information about the used kits. Please provide more information about the oligonucleotide primers used for the nested-PCR amplification and sequencing of the genomic region of HBV.
Divide your results into separate points, for example: Analysis of Resistance Mutations
The results from СLIA/ELISA are not presented. Please explain the purpose of each assay.
Please provide data on HBV DNA Quantification.
The quality of the figures is poor. Please provide better quality figures and try to group the results and reduce the number of figures
Can you provide one phylogenetic neighbor-joining tree of HBV sequences based on the alignment?
Author Response
For research article
|
Response to Reviewer 3 Comments
|
||
|
1. Summary |
|
|
|
Thank you very much for taking the time to review this manuscript. Please find the detailed responses below and the corresponding revisions/corrections highlighted/in track changes in the re-submitted files. |
||
|
2. Questions for General Evaluation |
Reviewer’s Evaluation |
Response and Revisions |
|
Does the introduction provide sufficient background and include all relevant references? |
Yes/Can be improved/Must be improved/Not applicable |
|
|
Are all the cited references relevant to the research? |
Yes/Can be improved/Must be improved/Not applicable |
|
|
Is the research design appropriate? |
Yes/Can be improved/Must be improved/Not applicable |
|
|
Are the methods adequately described? |
Yes/Can be improved/Must be improved/Not applicable |
|
|
Are the results clearly presented? |
Yes/Can be improved/Must be improved/Not applicable |
|
|
Are the conclusions supported by the results? |
Yes/Can be improved/Must be improved/Not applicable |
|
|
3. Point-by-point response to Comments and Suggestions for Authors |
||
|
Comments 1: Hepatitis B is a vaccine-preventable liver infection caused by the hepatitis B virus. It can cause long-lasting, acute or chronic liver damage. Eremin et al. provide valuable insights for updating the HBV molecular epidemiology in Belarus. However, the manuscript contains many omissions and needs serious revision. The abstract should be rewritten to include an introductory sentence that shows the importance of HBV. The reader can't even orientate himself on what the researched disease is. Overall, the introduction is well-written. In the introduction is good to describe the classification of HBV and the genome organization. At the end of the introduction, it is necessary to include the aim of your study and the period of your research. Response 1: In the introduction, we described the genome structure (main genes) of the hepatitis B virus and the current classification (by genotypes and their distribution). We also provided data on the prevalence of viral hepatitis in our country. The time frame of the study (2014-2023) is given in the Materials and Methods section. Comments 2: Materials and methods are not well described and do not provide complete information about the exodus carried out. Divide your materials and methods into separate points, describing the methods you used coherently and logically. Please distinguish between immunological studies and molecular studies. and provide more information about the used kits. Please provide more information about the oligonucleotide primers used for the nested-PCR amplification and sequencing of the genomic region of HBV. Divide your results into separate points, for example: Analysis of Resistance Mutations Response 2: We have divided the Materials and Methods sections into separate fragments. The determination of resistance mutations was not the main objective of our research. These are discussed in more detail in the article on the treatment of patients with chronic hepatitis B virus disease. Comments 3: The results from CMIA/ELISA are not presented. Please explain the purpose of each assay. Response 3: Positive samples were taken in the work, positive by both CMIA /ELISA and PCR. Some data on PCR we have given in the corresponding section of the article. Comments 4: Please provide data on HBV DNA Quantification. Response 3: We have provided some data on the quantification of HBV DNA in the PCR section. Comments 4: The quality of the figures is poor. Please provide better quality figures and try to group the results and reduce the number of figures Can you provide one phylogenetic neighbor-joining tree of HBV sequences based on the alignment? Response 4: The ML algorithm is optimal for processing a large number of nucleotide sequences and constructing a phylogenetic tree. We also used the neighbor joining method, but the ML algorithm is more representative. Unfortunately, the construction of phylogenetic trees from a large number of nucleotide sequences takes a lot of pages. For this reason, we have highlighted separate clusters with samples from our country in red.
|
||
|
|
||
|
|
||
|
|
||
|
4. Response to Comments on the Quality of English Language |
||
|
Point 1: |
||
|
Response 1: We have improved the English text and hope it is more readable.
|
||
|
5. Additional clarifications |
||
|
[Here, mention any other clarifications you would like to provide to the journal editor/reviewer.] |
||
HEPATITIS B VIRUS GENOTYPES AND SUBGENOTYPES CIRCULATING IN BELARUS
1Vladimir Eremin, 1Fedor Karpenko, 2Igor Karpov, 3Valeriy Semenov, 3Ina Oiestad
1Republican Scientific and Practical Center for Transfusiology and Medical Biotechnology
2Belarusian State Medical University, Department of Infectious Diseases, Belarus
3Vitebsk State Medical University, Department of Infectious Diseases, Belarus
Abstract. Approximately 800-900 new cases of chronic forms of hepatitis B are reported in Belarus annually. Compulsory vaccination, introduced to the country in the mid-90s, produces a certain positive effect on reducing the number of HBV cases. However, around 75,000 patients with chronic hepatitis B are estimated to live in the country at the moment. The main goal of this research was to determine the genotypes/subgenotypes of the hepatitis B virus and establish their role in maintaining the epidemic process in the country. Serological (CMIA, ELISA), molecular biological (PCR, sequencing) and bioinformatic (phylogenetic analysis) methods have been used to obtain results. As studies have shown, in 722 (81.7%) samples genotype D has been determined; in 156 (17.7%) - genotype A; in 3 (0.3%) - genotype C and only in 1 sample (0.1%) – genotype B. In two cases (0.2%) a recombinant form of the hepatitis B virus has been detected. The epidemic process of hepatitis B in the country is supported mainly by the circulation of “local” variants of the virus. At the same time, there are occasional introductions of new genetic variants from the outside of the country.
Keywords. HBV, genotypes/subgenotypes, sequencing, phylogenetic analysis, hepatitis B.
Introduction. Hepatitis B virus (HBV) infection remains a serious health concern despite the HBV vaccine introducing in the late 1990s. Approximately two billion cases of the disease were reported worldwide from the time of HBV discovery to the beginning of the 21st century. More than 400 million of them subsequently became chronic HBV carriers [1]. Chronic HBV infection is considered a major cause of many liver-related medical complications such as hepatocellular carcinoma, liver cirrhosis, and liver failure [2,3,4]. Due to the virus complexity, HBV genome contains an overlapping region that encodes four different genes (S, C, P and X). The S gene encodes exclusively the HBV surface envelope protein in its long open reading frame [6]. The presence of start codons allows the gene splicing into small, medium and large polypeptides, which are typically used either as a single target gene or in combination with other HBV genes to detect HBV DNA [5]. HBV core antigen is commonly encoded by the C gene, whereas the X gene encodes protein X [7]. The P gene encodes a polymerase protein, an integral component of the reverse transcription process during HBV replication. The HBV replication cycle lacks proofreading properties, resulting in progeny with a high genomic variability [5]. Due to significant genetic diversity, HBV is categorized into ten genotypes (A-J) with 7.5 % intergroup variation [8]. In addition to E and G, all genotypes are classified into 25 subgenotypes with 4% amino acid variability [9,10]. HBV genotypes are distributed differently depending on the geography: HBV-B, HBV-C and HBV-E are most common in Oceania and East Asia, while HBV-E is most prevalent in Central and West Africa. HBV-F and HBV H are found only in Alaska and Latin America. In contrast, HBV-D is a global pandemic. In Australia, Europe, Indonesia, North Africa and Western Asia, HBV-D1 is the most common virus, while HBV-D2 is found in Albania, Japan, Malaysia, North-Eastern Europe, Russia and the UK [11,12,13,14,15]. The progression and natural course of the disease varies among different HBV genotypes. These factors can complicate HBV infection treatment since the efficacy of known therapeutics turns out to be ineffective against certain genotypes and new genotypic variants [8,9]. Thus, there is a global exigency for constant update in genotypic information and surveys of HBV-infected patients [15]. The Republic of Belarus is a country with a moderate level of prevalence of parenteral viral hepatitis B. The inclusion of the vaccine against HBV infection into the Preventive Vaccination Schedule has made it possible to reduce the incidence of acute hepatitis B in the country by 6 times over the past 10 years and allows currently to consider Belarus as a country with a low level of prevalence of acute hepatitis B (less than 2% of the population). The most affected age groups are teenagers and adults 15 to 40 years old. The incidence rate in this specific age group is significantly higher than in other age groups of the population of the Republic. Approximately 800-900 new cases of chronic viral hepatitis B are detected annually in the country, despite a significant decrease of occurrence of acute viral hepatitis B. According to the epidemiological essays, 75,000 patients with chronic hepatitis B currwntly live in Belarus.Materials and methods Samples. 884 blood serum/plasma samples were tested for HBV markers with CMIA /ELISA and PCR methods. The samples were obtained from patients and blood donors with acute and chronic HBV from different regions of the country. The age of the patients ranged from 4 years to 78 years. 404 samples were obtained from males and 250 – from females. In 230 cases gender and age were not specified. Serum/plasma samples were collected and analyzed during 2014-2023. Serological tests. The commercial kit chemiluminescent microparticle immunoassay (CMIA) Architect HBsAg Qualitative II Reagent Kit, Abbott, USA was applied to detect HBsAg. Positive results were confirmed using the Architect HBsAg Qualitative II Confirmatory Reagent Kit, Abbott, USA as well as ELISA “Vectogep B-HBs antigen», manufactured by “Vector-Best”, Novosibirsk, Russia. PCR. The polymerase chain reaction was used to determine DNA of the hepatitis B virus using test systems, produced by Vector-Best CJSC: “RealBest HBV DNA (quantitative”, “RealBest HIV RNA (quantitative”) in accordance with the manufacturer’s instructions. Primers. We used the previously described by L Serfaty et al [16] primer pairs for sequencing the S and P regions of the hepatitis B virus genome. DNA isolation. Viral RNA/DNA from the blood serum/plasma samples were isolated using the “Kit of reagents for the isolation of NK” (manufactured by “Vector-Best”, Russia) and the “Kit of reagents for the isolation of RNA/DNA from clinical material” “RIBO- sorb", "RNA-prep", Russia. The manufacturers’ instructions were strictly followed. Nested PCR was performed using previously described primer pairs p1/pR5 (position 1197-1178) and p4/pR2 (1017-997) [16] according to the following protocol in a volume of 25µl: 2,5 µl 10х buffer+ MgCl2; 0,25 µl 25mM dNTPs; 0,5 µl 10 µM p1; 0,5 µl 10 µM pR5; 0,8 10 µM Taq polymerase; 18,45 µl bdH20. The first round of PCR was carried out according to the following protocol: one denaturation cycle at 95oC for 2 minutes and 30 cycles of amplification were performed with denaturation at 95°C for 30 sec, annealing at 53°C for 30 sec, and extension at 68°C for 1 minutes, then one cycle at 68oC for 5 minutes and storage at 10oC. In the second round of PCR, the annealing temperature was 50°C; the rest parameters were not changed. Amplified DNA fragments were analyzed on a 2% agarose gel. Electrophoresis was carried out at 10 w/cm of gel in TRIS-acetate buffer, pH 8.0. DNA was visualized using the Vitran Photo gel documentation system (Biocom Company LLC, Russia). The fragment size was determined according to the molecular weight of marker of 100-1000 bp (Fermentas, Lithuania).
The resulting DNA fragments were purified using the NimaGen ExS-Pure kit, Holland.
PCR sequencing was carried out using the second round primer pair p4/pR2 in the volume of 10 µl according to the following protocol: 5x seqbuf - 2 µl; BigDye Terminator v. 3.1 (applied biosystems) - 1 µl; p4/pR4 - 2 µl; bdH20 - 3 bdH20; DNA - 2 µl.
Electrophoresis of HBV DNA fragments obtained and purified after PCR sequencing was performed on AB 3500 genetic analyzer, USA. Phylogenetic analysis of the obtained HBV DNA fragments was conducted using the computer programs: Sequencing Analysis v.6, BioEdit, SeqScape v3, MEGAX and Genious 8.1. Phylogenetic trees were built using the ML (maximum likelihood) algorithm in the PHYML program. The SH-aLRT test was performed to calculate the statistical significance of the clusters. Clusters with a support node ≥0.9 were considered reliable. Mutations in the HBV genome in the S and P regions were determined using the programs: https://www.geno2pheno.org, http://www.hiv-grade.de/hbv_grade and https://hivdb.stanford.edu/HBV/HBVseq. Results The studies showed the following: 722 (81.7%) out of all 884 sequenced and analyzed samples, had genotype D; 156 (17.7%) - genotype A; 3 (0.3%) - genotype C and the only 1 (0.1%) – genotype B. For two cases (0.2%) a recombinant form of the hepatitis B virus was detected. Genotype D was found to be represented by subgenotypes - D1 (145/20.1%), D2 (371/51.4%), D3 (201/27, 8%) and D4 (5/0.7%); A – A2; C – C1 and C2; B – B4; recombinant forms C2/ D and A2/ С2 were detected (Figure 1). Figure 1. HBV subgenotypes identified in Belarus In 12 cases, the virus with mutation mutations to reverse transcriptase inhibitors Lamivudine, Zeffix® and Telbivudine, Tyzeka®, Sebivo® and partial resistance to Entecavir, Baraclude® was identified. Moreover, in 4 cases, resistance of mutations was identified in subgenotypes A2 and D2, in 3 cases- in the genome of subgenotype D1, and in one case - in subgenotype D3. Mutations were most often recorded at positions 180M and 204V - in 8 cases, 204I - in three cases, 80I and 173L - 2 times each, and 80V and 181T in one case (Table 1). Table 1 – Resistance of mutations identified in different HBV subgenotypes
|
N |
Subgenotypes |
Mutations |
|
1. |
А2 |
80V-1; 180M-3; 181T-1; 204V-3; |
|
2. |
D1 |
173L-1; 180M-3; 204V-3; |
|
3. |
D2 |
80I-2; 173L-1; 180M-2; 204V-2; 204I-2; |
|
4. |
D3 |
204I-1; |
Determining mutations is very important for infectious disease doctors, since substitutions at positions 180M, 181T, 204V and 204I lead to resistance to Lamivudine (Zeffix), Adefovir (Hepsera) and Telbivudine (Tyzeka, Sebivo) and require a change in treatment regimen.
Phylogenetic analysis of subtype A2 DNA sequences showed that most samples from Belarus were found nearby and formed at least 7 clusters (Figures 2, 3, 4). At the same time, some samples were in different groups and formed clusters with samples from the USA, Cuba, Brazil, similarly to Western European countries, mainly from Poland, Italy, France, Belgium, Germany and some others (Figures 2,3,4).
|
2 |
|
3 |
|
4 |
Fig.2, 3, 4 Phylogenetic analysis of subtype А2 (Here and in other figures, samples from Belarus are indicated in red)
12 clusters were identified while analyzing samples of subtype D1. Most of the sequenced and analyzed HBV DNA samples were identified with sequences from Pakistan, India, Iran, Bangladesh and some European countries such as France, Holland, Italy. Two large clusters, consisting mainly of samples from Belarus, were found with sequences from Iran and Italy, as well as Pakistan and France, Oman, Holland and Russia (Figures 5,6).
|
5 |
|
6 |
Fig. 5 and 6 Phylogenetic analysis of subtype D1
Sequences of subtype D2 formed 14 clusters, which were present in the samples from Belarus described earlier. They showed similarities with samples from Estonia, USA, India, Spain, Russia, Cameroon, Latvia, Iran and Belgium, Figure 7, 8, 9.
|
7 |
|
8 |
|
9 |
Fig. 7, 8, 9 Phylogenetic analysis of subtype D2
Phylogenetic analysis of D3 subtype showed that the samples formed 14 clusters mainly originated from India, Croatia, Russia, Italy, Germany and even Brazil. The largest cluster included samples from Belarus, with the previously described sequences. These clusters originated from Poland, Russia, Rwanda, Estonia and Albania, Figure 10.
Fig. 10 Phylogenetic analysis of subtype D3
The phylogenetic analysis of the subtype D4 samples showed that they all were found in the same group and clustered with HBV sequences from Cuba and Haiti, Figure 11.
Fig. 11 Phylogenetic analysis of subtype D4
Samples of subgenotypes C1 and C2 were identified in adult patients and a child from China. Sequences of subtype C1 formed a cluster, with samples from the USA and Vietnam, and subtype C2 with reference samples from South Korea, USA and China, Figure 12,13.
Fig .12, 13. Phylogenetic analysis of subtypes C1 and C2
The subtype B4 sample was revealed in a student from Vietnam who came to study in our country.
Finally, both samples with recombinant forms of HBV were identified in residents of Belarus
Discussion
Approximately 800-900 new cases of chronic forms of hepatitis B are reported in Belarus annually. Compulsory vaccination, introduced to the country in the mid-90s, produces a certain positive effect on reducing the number of HBV cases. However, around 75,000 patients with chronic hepatitis B are estimated to live in the country at the moment. Our previous study showed that subgenotype D2 (55.8%) was dominant in Belarus, followed by D3 (18,3%), D1 (11,6%) and A2 (11,6%) [17]. The conducted studies show that the structure of genotypes/subgenotypes in the country has actually remained unchanged. Genotype D still dominates in the country and accounts for 81.7% of all analyzed cases. Genotype D was found to be represented by subgenotypes D1 (145/20.1%), D2 (371/51.4%), D3 (201/27, 8%) and D4 (5/0.7%); A – A2; C – C2; B – B4; recombinant forms C2/ D and A2/ С2 were detected. Possible causes of the infection in patients were determined i.e. they occurred due to the circulation of “domestic” variants of the virus and/or the introduction of the virus from outside. The findings obtained show that the epidemic transmission of viral hepatitis B in the country is maintained mainly due to the circulation of local variants of the virus. It is confirmed by the examples of subgenotypes A2, D1, D2 and D3, where the largest clusters were formed by samples from Belarus. However, it should be noted that new variants of HBV are constantly being introduced into the country. The main routes of introduction of A2 subgenotype are Western Europe countries and even the USA, Cuba, Brazil; for D1 - Pakistan, India, Iran, Bangladesh, Italy, Holland and Russia; D2 - Estonia, USA, India, Spain, Russia, Cameroon, Latvia, Iran and Belgium; D3 - Poland, Russia, Rwanda, Estonia and Albania. All samples of D4 subgenotype were clustered with sequences from Cuba and Haiti, C1 and C2 – from China, and B4 – from Vietnam. Two recombinant forms C2/D and A2/D were identified in the residents of our country. Comparing the distribution of different HBV genotypes in neighboring countries and even in different regions of the same country, we may conclude that they vary. For example, genotype D significantly dominates in the European part of Russian Federation, while, for example, in Yakutia, genotype A occupies one of the leading positions (36.4% - A and 58,6% - D) [18]. In Asia, particularly in its eastern part, genotypes B and C dominate, and the C2 subgenotype is epidemic in China [19, 20]. In Western Europe, in particular in Portugal HBV genotype A (HBV/A) was the most prevalent genotype (41.5%), followed by D [HBV/D; (33.8%)], and E [HBV/E; (24.6%)]. Subgenotypes A1 (HBV/A1) and D4 (HBV/D4) were almost equally prevalent with 23.1% and 22.3%, respectively, followed by A2 (HBV/A2) with 16.2% and D3 (HBV/D3) with 11.5% [21].
The Republic of Belarus is geographically located in the center of Europe. The residents of the country are actively travelling and many people from other countries relocate to Belarus for work and study. This fact can easily explain the introduction of the new variants of HBV to the Republic.
The results of this study contribute valuable information of the molecular genetic characteristics of HBV, prevailing in the Republic of Belarus. Possible routes of the introduction of the new variants of HBV have been identified. The main mutations in the P region of the virus genome leading to resistance to antiviral drugs have detected. Identification of the mutations at positions 181T, 204V, 204I, which determine resistance not only to Lamivudine (Zefffix), but also to Adefovir (Hepsera) and Telbivudine (Tyzeka, Sebivo), made it possible to use Tenofovir DF for treatment of the patients. The compensatory mutations 80V, 173L, 80I have been found in the complex with the mutations leading to virus resistance to antiviral drugs.
Conclusions
To summarize, should be noted that the use of the methods of molecular epidemiology solve many issues for both: epidemiologists and infectious diseases specialists. This knowledge is essential to plan preventive measures for the directions and routes of its’ spread in the country by determining the genotypes/subgenotypes of the hepatitis B virus. The quality of living of our patients could be improved by prescribing new treatment regimens, based on the resistance mutations identification.
REFERENCES
- Thun M.J., DeLancey J.O., Center M.M., Jemal A., Ward E.M. The global burden of cancer: Priorities for prevention. Carcinogenesis. 2010;31:100–110.
- Philips C.A., Ahamed R., Abduljaleel J.K., Rajesh S., Augustine P. Critical Updates on Chronic Hepatitis B Virus Infection in 2021. Cureus. 2021;13:e19152.
- Kao J.-H., Chen P.-J., Lai M.-Y., Chen D.-S. Hepatitis B genotypes correlate with clinical outcomes in patients with chronic hepatitis B. Gastroenterology. 2000;118:554–559.
- Aghakhani A., Hamkar R., Zamani N., Eslamifar A., Banifazl M., Saadat A., Sofian M., Adibi L., Irani N., Mehryar M. Hepatitis B virus genotype in Iranian patients with hepatocellular carcinoma. Int. J. Infect. Dis. 2009;13:685–689.
- Beck J., Nassal M. Hepatitis B virus replication. World J. Gastroenterol. 2007;13:48–64.
- Pollicino T., Cacciola I., Saffioti F., Raimondo G. Hepatitis B virus PreS/S gene variants: Pathobiology and clinical implications. J. Hepatol. 2014;61:408–417.
- Ie S.I., Thedja M.D., Roni M., Muljono D.H. Prediction of conformational changes by single mutation in the hepatitis B virus surface antigen (HBsAg) identified in HBsAg-negative blood donors. Virol. J. 2010;7:326.
- Lin C.-L., Kao J.-H. The clinical implications of hepatitis B virus genotype: Recent advances. J. Gastroenterol. Hepatol. 2011;26((Suppl. 1)):123–130.
- Zhu S.S., Zhang H.F., Wang H.B., Dong Y., Chen D., Jia W., Gan Y., Chen J. Relation of viral genotypes to clinical features in children with chronic hepatitis B. Chin. J. Exp. Clin. Virol. 2008;22:192–194.
- Shamseer L., Moher D., Clarke M., Ghersi D., Liberati A., Petticrew M., Shekelle P., Stewart L.A. Preferred reporting items for systematic review and meta-analysis protocols (PRISMA-P) 2015: Elaboration and explanation. BMJ. 2015;349:g7647.
- George B.J., Aban I.B. An application of meta-analysis based on DerSimonian and Laird method. J. Nucl. Cardiol. 2016;23:690–692.
- Fletcher J. What is heterogeneity and is it important? BMJ. 2007;334:94–96.
- Higgins J.P.T., Thompson S.G., Deeks J.J., Altman D.G. Measuring inconsistency in meta-analyses. BMJ. 2003;327:557–560.
- Wallace B.C., Dahabreh I.J., Trikalinos T.A., Lau J., Trow P., Schmid C.H. Closing the Gap between Methodologists and End-Users: R as a Computational Back-End. J. Stat. Softw. 2012;49:2–15.
- Munn Z., MClinSc S.M., Lisy K., Riitano D., Tufanaru C. Methodological guidance for systematic reviews of observational epidemiological studies reporting prevalence and cumulative incidence data. Int. J. Evid. Based Healthc. 2015;13:147–153.
- Serfaty L., Thabut D., Zoulim F., Andreani T., Eres O. C., Carbonell N., Loria A., and Poupon R. / Hepatology – 2001 – V.34 – N.- 3 – P.573-577.
- Olinger C.M., Lazouskaya N.V., Eremin V.F., Muller C.P. Multiple genotypes of hepatitis B and C viruses in Belarus: similarities with Russia and European influences. Clin Microbiol Infect – 2008; 14:575-581.
- Anastasia A Karlsen, Karen K Kyuregyan, Olga V Isaeva, Vera S Kichatova2, Fedor A Asadi Mobarkhan, Lyudmila V Bezuglova, Irina G Netesova, Victor A Manuylov, Andrey A Pochtovyi, Vladimir A Gushchin, Snezhana S Sleptsova, Margarita E Ignateva, Mikhail I Mikhailov Different evolutionary dynamics of hepatitis B virus genotypes A and D, and hepatitis D virus genotypes 1 and 2 in an endemic area of Yakutia, Russia // /BMC Infect Dis – 2022; 12;22:452
- Kizito Eneye Bello, Tuan Nur Akmalina Mat Jusoh, Ahmad Adebayo Irekeola, Norhidayah Abu, Nur Amalin Zahirah Mohd Amin, Nazri Mustaffa, Rafidah Hanim Shueb/ A Recent Prevalence of Hepatitis B Virus (HBV) Genotypes and Subtypes in Asia: A Systematic Review and Meta-Analysis // Healthcare (Basel) – 2023; 1;11(7):1011.
- Bing Sun, Aida Andrades Valtueña, Arthur Kocher, Shizhu Gao, Chunxiang Li, Shuang Fu, Fan Zhang, Pengcheng Ma, Xuan Yang, Yulan Qiu, Quanchao Zhang, Jian Ma, Shan Chen, Xiaoming Xiao, Sodnomjamts Damchaabadgar, Fajun Li, Alexey Kovalev, Chunbai Hu, Xianglong Chen, Lixin Wang, Wenying Li, Yawei Zhou, Hong Zhu Johannes Krause, Alexander Herbig, Yinqiu Cui/ Origin and dispersal history of Hepatitis B virus in Eastern Eurasia // Nat Commun, 2024; 5;15:2951.
- Rute Marcelino, Ifeanyi Jude Ezeonwumelu, André Janeiro, Paula Mimoso, Sónia Matos, Veronica Briz, Victor Pimentel, Marta Pingarilho, Rui Tato Marinho, José Maria Marcelino, Nuno Taveira, Ana Abecasis / . Phylogeography of hepatitis B virus: The role of Portugal in the early dissemination of HBVworldwide // PLoS One. 2022; 17(12):
Acknowledgments
We would like to show our gratitude to the Abbott Transfusion Medicine Medical affairs team for financial support of the publication.
Abstract. Annually 800-900 new cases of hepatitis B chronic forms are registered in Belarus. Compulsory vaccination introduced in the mid-90s certainly produces a positive effect on reducing the number of HBV cases. About 75,000 patients with chronic hepatitis B living in the country are estimated to live in the country at the moment. The main goal of this work was to determine the genotypes/subgenotypes of the hepatitis B virus and establish their role in maintaining the epidemiological process on the territory of Belarus. Serological (CMIA, ELISA), molecular biological (PCR, sequencing) and bioinformatic (phylogenetic analysis) methods have been used. As studies have shown, out of all 884 sequenced and analyzed samples, 722 (81.7%) of them have genotype D; 156 (17.7%) - genotype A; 3 (0.3%) - genotype C and the only 1 (0.1%) – genotype B. For two cases (0.2%) a recombinant form of the hepatitis B virus was detected. The epidemic process of hepatitis B in the country is supported mainly by the circulation of “local” variants of the virus. At the same time, there are occasional introductions of new variants from outside the country.
Keywords. HBV, genotypes/subgenotypes, sequencing, phylogenetic analysis.
Round 2
Reviewer 1 Report
Comments and Suggestions for Authors
Although the authors have addressed most of my concerns and comments in their revised manuscript. However, there are still several points needs to be noted.
1. To my previous comment 5: You may want to show the node labels while not the branch labels in your phylogenetic trees. In the methods, it is written “Clusters with a support node ≥0.9 were considered reliable.” Additionally, please indicate the bootstrap methods in your analysis.2
2. To my previous comment 6: The GenBank accession numbers should be at least indicated in the manuscript. Otherwise, how can the readers retrieve them?
3. Figures 2 to 13: The presentations of phylogenetic trees are incomplete. You may collapse some branches which are less important. Is there any specific meaning for the background using different colors in these figures?
4. I doubt the current conclusions is associated with the findings presented in this study.
Author Response
Rev_1
Comments and Suggestions for Authors
Although the authors have addressed most of my concerns and comments in their revised manuscript. However, there are still several points needs to be noted.
- To my previous comment 5: You may want to show the node labels while not the branch labels in your phylogenetic trees. In the methods, it is written “Clusters with a support node ≥0.9 were considered reliable.” Additionally, please indicate the bootstrap methods in your analysis.2
Phylogenetic trees were constructed using the ML (maximum likelihood) algorithm in PHYML (Phylogenetic maximum likelihood) and GARLY (Genetic Algorithm for Rapid Likelihood Inference) programmes, with the nucleotide substitution model GTR+I+G (I-proportion of invariable sites, G - Gamma shape parameter). Tree topology optimisation - Best of NNIs and SPRs. The SH-alRT test was used to calculate the statistical reliability of clusters. Clusters of the phylogenetic tree whose root branch had a value ≥0.9 were considered reliable. The phylogenetic trees were visualised using FigTree v.4.2 software.
Yes, indeed, clusters with support node ≥0.9 were considered reliable. However, this is more the case for deciphering blood-borne infections or hepatitis B outbreaks among drug addicts and so on. In our case, the objective of the study was to identify possible directions of virus introduction into the country and to determine at the expense of which virus variants the epidemiological process in the country is supported.
- To my previous comment 6: The GenBank accession numbers should be at least indicated in the manuscript. Otherwise, how can the readers retrieve them?
We have registered more than 800 nucleotide sequences of HIV, HCV and HBV in GenBank. We are currently registering HBV sequences collected in recent years and linking to the article and journal in which they will be presented. If we do it the other way round, it will take a long time and the article will be published much later.
- Figures 2 to 13: The presentations of phylogenetic trees are incomplete. You may collapse some branches which are less important. Is there any specific meaning for the background using different colors in these figures?
We have large experimental material and, consequently, large phylogenetic trees. In this regard, we have identified clusters that allow the reader to obtain information about the reasons for the maintenance of the epidemiological process in our country and to identify possible directions of virus introduction into the country. It is clear that the support nodes ≥0.9 will be between samples from our country (family cases - they are present on the trees), and between our samples and reference sequences from GenBank the support nodes will be smaller. In the next publication, we will present data on deciphering family cases, outbreaks in health care facilities, and blood borne infections.
- I doubt the current conclusions is associated with the findings presented in this study.
I cannot agree with your remark, as molecular epidemiology methods really allow us to solve many issues and improve the epidemiological situation, and in terms of timely change of therapy regimens, as well as to determine the prognosis of the course of the disease (we do it by sequencing the C region of the genome), etc.
HEPATITIS B VIRUS GENOTYPES AND SUBGENOTYPES CIRCULATING IN BELARUS
1Vladimir Eremin, 1Fedor Karpenko, 2Igor Karpov, 3Valeriy Semenov, 3Ina Oiestad
1Republican Scientific and Practical Center for Transfusiology and Medical Biotechnology
2Belarusian State Medical University, Department of Infectious Diseases, Belarus
3Vitebsk State Medical University, Department of Infectious Diseases, Belarus
Abstract. Approximately 800-900 new cases of chronic forms of hepatitis B are reported in Belarus annually. Compulsory vaccination, introduced to the country in the mid-90s, produces a certain positive effect on reducing the number of HBV cases. However, around 75,000 patients with chronic hepatitis B are estimated to live in the country at the moment. The main goal of this research was to determine the genotypes/subgenotypes of the hepatitis B virus and establish their role in maintaining the epidemic process in the country. Serological (CMIA, ELISA), molecular biological (PCR, sequencing) and bioinformatic (phylogenetic analysis) methods have been used to obtain results. As studies have shown, in 722 (81.7%) samples genotype D has been determined; in 156 (17.7%) - genotype A; in 3 (0.3%) - genotype C and only in 1 sample (0.1%) – genotype B. In two cases (0.2%) a recombinant form of the hepatitis B virus has been detected. The epidemic process of hepatitis B in the country is supported mainly by the circulation of “local” variants of the virus. At the same time, there are occasional introductions of new genetic variants from the outside of the country.
Keywords. HBV, genotypes/subgenotypes, sequencing, phylogenetic analysis, hepatitis B.
Introduction. Hepatitis B virus (HBV) infection remains a serious health concern despite the HBV vaccine introducing in the late 1990s. Approximately two billion cases of the disease were reported worldwide from the time of HBV discovery to the beginning of the 21st century. More than 400 million of them subsequently became chronic HBV carriers [1]. Chronic HBV infection is considered a major cause of many liver-related medical complications such as hepatocellular carcinoma, liver cirrhosis, and liver failure [2,3,4]. Hepatitis B virus, like HIV, has three main transmission mechanisms: parenteral (through blood and its products and/or co-injection of drugs), vertical (from infected mother to child) and sexual transmission. HBV belongs to the Hepadnaviridae family, with a genome of approximately 3.2 kb in length. Due to the virus complexity, HBV genome contains an overlapping region that encodes four different genes (S, C, P and X). The S gene encodes exclusively the HBV surface envelope protein in its long open reading frame [6]. The presence of start codons allows the gene splicing into small, medium and large polypeptides, which are typically used either as a single target gene or in combination with other HBV genes to detect HBV DNA [5]. HBV core antigen is commonly encoded by the C gene, whereas the X gene encodes protein X [7]. The P gene encodes a polymerase protein, an integral component of the reverse transcription process during HBV replication. The HBV replication cycle lacks proofreading properties, resulting in progeny with a high genomic variability [5].
Due to significant genetic diversity, HBV is categorized into ten genotypes (A-J) with 7.5 % intergroup variation [8]. In addition to E and G, all genotypes are classified into 25 subgenotypes with 4% amino acid variability [9,10]. HBV genotypes are distributed differently depending on the geography: HBV-B, HBV-C and HBV-E are most common in Oceania and East Asia, while HBV-E is most prevalent in Central and West Africa. HBV-F and HBV H are found only in Alaska and Latin America. In contrast, HBV-D is a global pandemic. In Australia, Europe, Indonesia, North Africa and Western Asia, HBV-D1 is the most common virus, while HBV-D2 is found in Albania, Japan, Malaysia, North-Eastern Europe, Russia and the UK [11,12,13,14,15]. Now the HBV is divided into ten genotypes (A-J) and 24 subgenotypes (A1–A3, B1–B5, C1–C6, D1–D6 and F1–F4. A systematic genotype and subgenotype re-ranking of hepatitis B virus under a novel classification standard [16].
The progression and natural course of the disease varies among different HBV genotypes. These factors can complicate HBV infection treatment since the efficacy of known therapeutics turns out to be ineffective against certain genotypes and new genotypic variants [8,9]. Thus, there is a global exigency for constant update in genotypic information and surveys of HBV-infected patients [15].
In Belarus, selective hepatitis B vaccination for epidemic indications has been organised since 1996 and included in the national preventive vaccination calendar since 2000. Vaccination of newborns, children aged 13 years and certain risk groups (health care workers and persons in domestic contact with infected persons) has helped to reduce the incidence of hepatitis B infection. Vaccination against viral hepatitis B has made it possible to reduce the incidence of acute hepatitis B almost six-fold over the past 10 years (from 5.9 to 1.02 per 100,000 population) and to consider Belarus as a country with a low incidence of acute viral hepatitis B (less than 2 per cent of the population). This means that vaccination coverage against hepatitis B has been over 97 percent for several decades. This is a good result. In 2022, a WHO-supported study of the child population in Belarus showed how many of those vaccinated had HBsAg, the surface antigen of viral hepatitis B: - It was detected in only one child. The most affected age groups are teenagers and adults 15 to 40 years old. The incidence rate in this specific age group is significantly higher than in other age groups of the population of the Republic. Approximately 800-900 new cases of chronic viral hepatitis B are detected annually in the country, despite a significant decrease of occurrence of acute viral hepatitis B. According to the epidemiological essays, 75,000 patients with chronic hepatitis B currently live in Belarus.
Materials and methods Samples. 884 blood serum/plasma samples were tested for HBV markers with CMIA /ELISA and PCR methods. The samples were obtained from patients and blood donors with acute and chronic HBV from different regions of the country. The age of the patients ranged from 4 years to 78 years. 404 samples were obtained from males and 250 – from females. In 230 cases gender and age were not specified. Serum/plasma samples were collected and analyzed during 2014-2023. Serological tests. The commercial kit chemiluminescent microparticle immunoassay (CMIA) Architect HBsAg Qualitative II Reagent Kit, Abbott, USA was applied to detect HBsAg. Positive results were confirmed using the Architect HBsAg Qualitative II Confirmatory Reagent Kit, Abbott, USA as well as ELISA “Vectogep B-HBs antigen», manufactured by “Vector-Best”, Novosibirsk, Russia.
PCR. The polymerase chain reaction was used to determine DNA of the hepatitis B virus using test systems, produced by Vector-Best CJSC: “RealBest HBV DNA (quantitative”, “RealBest HIV RNA (quantitative”) in accordance with the manufacturer’s instructions. Viral load in the samples ranged from 1.2x102 to 3.4x106 IU/ml HBV DNA.
Primers. We used the previously described by L Serfaty et al [17] primer pairs for sequencing the S and P regions of the hepatitis B virus genome. DNA isolation. Viral RNA/DNA from the blood serum/plasma samples were isolated using the “Kit of reagents for the isolation of NK” (manufactured by “Vector-Best”, Russia) and the “Kit of reagents for the isolation of RNA/DNA from clinical material” “RIBO- sorb", "RNA-prep", Russia. The manufacturers’ instructions were strictly followed. Nested PCR was performed using previously described primer pairs p1/pR5 (p1: 5-CCTGCTGGTGGCTCCAGTTC-3 at nucleotide position 55-76, and primer pR5: 5-GGT TGC GTC AGC AAA CAC TTG-3 at position 1197-1178) and p4/pR2 (p4 5-CTC ACA ATA CCG CAG AGT CTA GAC T-3 at nucleotide position 230-254, and pR2: 5-AAA GCC CAA AAG ACC CAC AAT-3 at position 1017-997) [17] according to the following protocol in a volume of 25µl: 2,5 µl 10х buffer+ MgCl2; 0,25 µl 25mM dNTPs; 0,5 µl 10 µM p1; 0,5 µl 10 µM pR5; 0,8 10 µM Taq polymerase; 18,45 µl bdH20. The first round of PCR was carried out according to the following protocol: one denaturation cycle at 95oC for 2 minutes and 30 cycles of amplification were performed with denaturation at 95°C for 30 sec, annealing at 53°C for 30 sec, and extension at 68°C for 1 minutes, then one cycle at 68oC for 5 minutes and storage at 10oC. In the second round of PCR, the annealing temperature was 50°C; the rest parameters were not changed. As a result, we obtained a fragment of about 900 base pairs Amplified DNA fragments were analyzed on a 2% agarose gel. Electrophoresis was carried out at 10 w/cm of gel in TRIS-acetate buffer, pH 8.0. DNA was visualized using the Vitran Photo gel documentation system (Biocom Company LLC, Russia). The fragment size was determined according to the molecular weight of marker of 100-1000 bp (Fermentas, Lithuania).
The resulting DNA fragments were purified using the NimaGen ExS-Pure kit, Holland.
PCR sequencing was carried out using the second round primer pair p4/pR2 in the volume of 10 µl according to the following protocol: 5x seqbuf - 2 µl; BigDye Terminator v. 3.1 (applied biosystems) - 1 µl; p4/pR4 - 2 µl; bdH20 - 3 bdH20; DNA - 2 µl.
Electrophoresis of HBV DNA fragments obtained and purified after PCR sequencing was performed on AB 3500 genetic analyzer, USA. Phylogenetic analysis of the obtained HBV DNA fragments was conducted using the computer programs: Sequencing Analysis v.6, BioEdit, SeqScape v3, MEGAX and Genious 8.1. Phylogenetic trees were built using the ML (maximum likelihood) algorithm in the PHYML program. The SH-aLRT test was performed to calculate the statistical significance of the clusters. Clusters with a support node ≥0.9 were considered reliable. Mutations in the HBV genome in the S and P regions were determined using the programs: https://www.geno2pheno.org, http://www.hiv-grade.de/hbv_grade and https://hivdb.stanford.edu/HBV/HBVseq. Results The studies showed the following: 722 (81.7%) out of all 884 sequenced and analyzed samples, had genotype D; 156 (17.7%) - genotype A; 3 (0.3%) - genotype C and the only 1 (0.1%) – genotype B. For two cases (0.2%) a recombinant form of the hepatitis B virus was detected. Genotype D was found to be represented by subgenotypes - D1 (145/20.1%), D2 (371/51.4%), D3 (201/27, 8%) and D4 (5/0.7%); A – A2; C – C1 and C2; B – B4; recombinant forms C2/ D and A2/ С2 were detected (Figure 1). Figure 1. HBV subgenotypes identified in Belarus In 12 cases, the virus with mutation mutations to reverse transcriptase inhibitors Lamivudine, Zeffix® and Telbivudine, Tyzeka®, Sebivo® and partial resistance to Entecavir, Baraclude® was identified. Moreover, in 4 cases, resistance of mutations was identified in subgenotypes A2 and D2, in 3 cases- in the genome of subgenotype D1, and in one case - in subgenotype D3. Mutations were most often recorded at positions 180M and 204V - in 8 cases, 204I - in three cases, 80I and 173L - 2 times each, and 80V and 181T in one case (Table 1). Table 1 – Resistance of mutations identified in different HBV subgenotypes
|
N |
Subgenotypes |
Mutations |
|
1. |
А2 |
80V-1; 180M-3; 181T-1; 204V-3; |
|
2. |
D1 |
173L-1; 180M-3; 204V-3; |
|
3. |
D2 |
80I-2; 173L-1; 180M-2; 204V-2; 204I-2; |
|
4. |
D3 |
204I-1; |
Determining mutations is very important for infectious disease doctors, since substitutions at positions 180M, 181T, 204V and 204I lead to resistance to Lamivudine (Zeffix), Adefovir (Hepsera) and Telbivudine (Tyzeka, Sebivo) and require a change in treatment regimen.
Phylogenetic analysis of subtype A2 DNA sequences showed that most samples from Belarus were found nearby and formed at least 7 clusters (Figures 2, 3, 4). At the same time, some samples were in different groups and formed clusters with samples from the USA, Cuba, Brazil, similarly to Western European countries, mainly from Poland, Italy, France, Belgium, Germany and some others (Figures 2,3,4).
|
2 |
|
3 |
|
4 |
Fig.2, 3, 4 Phylogenetic analysis of subtype А2 (Here and in other figures, samples from Belarus are indicated in red)
12 clusters were identified while analyzing samples of subtype D1. Most of the sequenced and analyzed HBV DNA samples were identified with sequences from Pakistan, India, Iran, Bangladesh and some European countries such as France, Holland, Italy. Two large clusters, consisting mainly of samples from Belarus, were found with sequences from Iran and Italy, as well as Pakistan and France, Oman, Holland and Russia (Figures 5,6).
|
5 |
|
6 |
Fig. 5 and 6 Phylogenetic analysis of subtype D1
Sequences of subtype D2 formed 14 clusters, which were present in the samples from Belarus described earlier. They showed similarities with samples from Estonia, USA, India, Spain, Russia, Cameroon, Latvia, Iran and Belgium, Figure 7, 8, 9.
|
7 |
|
8 |
|
9 |
Fig. 7, 8, 9 Phylogenetic analysis of subtype D2
Phylogenetic analysis of D3 subtype showed that the samples formed 14 clusters mainly originated from India, Croatia, Russia, Italy, Germany and even Brazil. The largest cluster included samples from Belarus, with the previously described sequences. These clusters originated from Poland, Russia, Rwanda, Estonia and Albania, Figure 10.
Fig. 10 Phylogenetic analysis of subtype D3
The phylogenetic analysis of the subtype D4 samples showed that they all were found in the same group and clustered with HBV sequences from Cuba and Haiti, Figure 11.
Fig. 11 Phylogenetic analysis of subtype D4
Samples of subgenotypes C1 and C2 were identified in adult patients and a child from China. Sequences of subtype C1 formed a cluster, with samples from the USA and Vietnam, and subtype C2 with reference samples from South Korea, USA and China, Figure 12,13.
Fig .12, 13. Phylogenetic analysis of subtypes C1 and C2
The subtype B4 sample was revealed in a student from Vietnam who came to study in our country.
Finally, both samples with recombinant forms of HBV were identified in residents of Belarus
Discussion
Approximately 800-900 new cases of chronic forms of hepatitis B are reported in Belarus annually. Compulsory vaccination, introduced to the country in the mid-90s, produces a certain positive effect on reducing the number of HBV cases. However, around 75,000 patients with chronic hepatitis B are estimated to live in the country at the moment. Our previous study showed that subgenotype D2 (55.8%) was dominant in Belarus, followed by D3 (18,3%), D1 (11,6%) and A2 (11,6%) [18]. The conducted studies show that the structure of genotypes/subgenotypes in the country has actually remained unchanged. Genotype D still dominates in the country and accounts for 81.7% of all analyzed cases. Genotype D was found to be represented by subgenotypes D1 (145/20.1%), D2 (371/51.4%), D3 (201/27, 8%) and D4 (5/0.7%); A – A2; C – C2; B – B4; recombinant forms C2/ D and A2/ С2 were detected. Possible causes of the infection in patients were determined i.e. they occurred due to the circulation of “domestic” variants of the virus and/or the introduction of the virus from outside. The findings obtained show that the epidemic transmission of viral hepatitis B in the country is maintained mainly due to the circulation of local variants of the virus. It is confirmed by the examples of subgenotypes A2, D1, D2 and D3, where the largest clusters were formed by samples from Belarus. However, it should be noted that new variants of HBV are constantly being introduced into the country. The main routes of introduction of A2 subgenotype are Western Europe countries and even the USA, Cuba, Brazil; for D1 - Pakistan, India, Iran, Bangladesh, Italy, Holland and Russia; D2 - Estonia, USA, India, Spain, Russia, Cameroon, Latvia, Iran and Belgium; D3 - Poland, Russia, Rwanda, Estonia and Albania. All samples of D4 subgenotype were clustered with sequences from Cuba and Haiti, C1 and C2 – from China, and B4 – from Vietnam. Two recombinant forms C2/D and A2/D were identified in the residents of our country. Comparing the distribution of different HBV genotypes in neighboring countries and even in different regions of the same country, we may conclude that they vary. For example, genotype D significantly dominates in the European part of Russian Federation, while, for example, in Yakutia, genotype A occupies one of the leading positions (36.4% - A and 58,6% - D) [19]. In Asia, particularly in its eastern part, genotypes B and C dominate, and the C2 subgenotype is epidemic in China [20, 21]. In Western Europe, in particular in Portugal HBV genotype A (HBV/A) was the most prevalent genotype (41.5%), followed by D [HBV/D; (33.8%)], and E [HBV/E; (24.6%)]. Subgenotypes A1 (HBV/A1) and D4 (HBV/D4) were almost equally prevalent with 23.1% and 22.3%, respectively, followed by A2 (HBV/A2) with 16.2% and D3 (HBV/D3) with 11.5% [22].
The Republic of Belarus is geographically located in the center of Europe. The residents of the country are actively travelling and many people from other countries relocate to Belarus for work and study. This fact can easily explain the introduction of the new variants of HBV to the Republic.
The results of this study contribute valuable information of the molecular genetic characteristics of HBV, prevailing in the Republic of Belarus. Possible routes of the introduction of the new variants of HBV have been identified. The main mutations in the P region of the virus genome leading to resistance to antiviral drugs have detected. Identification of the mutations at positions 181T, 204V, 204I, which determine resistance not only to Lamivudine (Zefffix), but also to Adefovir (Hepsera) and Telbivudine (Tyzeka, Sebivo), made it possible to use Tenofovir DF for treatment of the patients. The compensatory mutations 80V, 173L, 80I have been found in the complex with the mutations leading to virus resistance to antiviral drugs.
Conclusions
To summarize, should be noted that the use of the methods of molecular epidemiology solve many issues for both: epidemiologists and infectious diseases specialists. This knowledge is essential to plan preventive measures for the directions and routes of its’ spread in the country by determining the genotypes/subgenotypes of the hepatitis B virus. The quality of living of our patients could be improved by prescribing new treatment regimens, based on the resistance mutations identification.
REFERENCES
- Thun M.J., DeLancey J.O., Center M.M., Jemal A., Ward E.M. The global burden of cancer: Priorities for prevention. Carcinogenesis. 2010;31:100–110.
- Philips C.A., Ahamed R., Abduljaleel J.K., Rajesh S., Augustine P. Critical Updates on Chronic Hepatitis B Virus Infection in 2021. Cureus. 2021;13:e19152.
- Kao J.-H., Chen P.-J., Lai M.-Y., Chen D.-S. Hepatitis B genotypes correlate with clinical outcomes in patients with chronic hepatitis B. Gastroenterology. 2000;118:554–559.
- Aghakhani A., Hamkar R., Zamani N., Eslamifar A., Banifazl M., Saadat A., Sofian M., Adibi L., Irani N., Mehryar M. Hepatitis B virus genotype in Iranian patients with hepatocellular carcinoma. Int. J. Infect. Dis. 2009;13:685–689.
- Beck J., Nassal M. Hepatitis B virus replication. World J. Gastroenterol. 2007;13:48–64.
- Pollicino T., Cacciola I., Saffioti F., Raimondo G. Hepatitis B virus PreS/S gene variants: Pathobiology and clinical implications. J. Hepatol. 2014;61:408–417.
- Ie S.I., Thedja M.D., Roni M., Muljono D.H. Prediction of conformational changes by single mutation in the hepatitis B virus surface antigen (HBsAg) identified in HBsAg-negative blood donors. Virol. J. 2010;7:326.
- Lin C.-L., Kao J.-H. The clinical implications of hepatitis B virus genotype: Recent advances. J. Gastroenterol. Hepatol. 2011;26((Suppl. 1)):123–130.
- Zhu S.S., Zhang H.F., Wang H.B., Dong Y., Chen D., Jia W., Gan Y., Chen J. Relation of viral genotypes to clinical features in children with chronic hepatitis B. Chin. J. Exp. Clin. Virol. 2008;22:192–194.
- Shamseer L., Moher D., Clarke M., Ghersi D., Liberati A., Petticrew M., Shekelle P., Stewart L.A. Preferred reporting items for systematic review and meta-analysis protocols (PRISMA-P) 2015: Elaboration and explanation. BMJ. 2015;349:g7647.
- George B.J., Aban I.B. An application of meta-analysis based on DerSimonian and Laird method. J. Nucl. Cardiol. 2016;23:690–692.
- Fletcher J. What is heterogeneity and is it important? BMJ. 2007;334:94–96.
- Higgins J.P.T., Thompson S.G., Deeks J.J., Altman D.G. Measuring inconsistency in meta-analyses. BMJ. 2003;327:557–560.
- Wallace B.C., Dahabreh I.J., Trikalinos T.A., Lau J., Trow P., Schmid C.H. Closing the Gap between Methodologists and End-Users: R as a Computational Back-End. J. Stat. Softw. 2012;49:2–15.
- Munn Z., MClinSc S.M., Lisy K., Riitano D., Tufanaru C. Methodological guidance for systematic reviews of observational epidemiological studies reporting prevalence and cumulative incidence data. Int. J. Evid. Based Healthc. 2015;13:147–153.
- Yin Y, He K, Wu B, Xu M, Du L, Liu W, Liao P, Liu Y, He M. /Heliyon - 2019 Oct 23;5(10):e02556.
- Serfaty L., Thabut D., Zoulim F., Andreani T., Eres O. C., Carbonell N., Loria A., and Poupon R. / Hepatology – 2001 – V.34 – N.- 3 – P.573-577.
- Olinger C.M., Lazouskaya N.V., Eremin V.F., Muller C.P. Multiple genotypes of hepatitis B and C viruses in Belarus: similarities with Russia and European influences. Clin Microbiol Infect – 2008; 14:575-581.
- Anastasia A Karlsen, Karen K Kyuregyan, Olga V Isaeva, Vera S Kichatova, Fedor A Asadi Mobarkhan, Lyudmila V Bezuglova, Irina G Netesova, Victor A Manuylov, Andrey A Pochtovyi, Vladimir A Gushchin, Snezhana S Sleptsova, Margarita E Ignateva, Mikhail I Mikhailov Different evolutionary dynamics of hepatitis B virus genotypes A and D, and hepatitis D virus genotypes 1 and 2 in an endemic area of Yakutia, Russia // /BMC Infect Dis – 2022; 12;22:452
- Kizito Eneye Bello, Tuan Nur Akmalina Mat Jusoh, Ahmad Adebayo Irekeola, Norhidayah Abu, Nur Amalin Zahirah Mohd Amin, Nazri Mustaffa, Rafidah Hanim Shueb/ A Recent Prevalence of Hepatitis B Virus (HBV) Genotypes and Subtypes in Asia: A Systematic Review and Meta-Analysis // Healthcare (Basel) – 2023; 1;11(7):1011.
- Bing Sun, Aida Andrades Valtueña, Arthur Kocher, Shizhu Gao, Chunxiang Li, Shuang Fu, Fan Zhang, Pengcheng Ma, Xuan Yang, Yulan Qiu, Quanchao Zhang, Jian Ma, Shan Chen, Xiaoming Xiao, Sodnomjamts Damchaabadgar, Fajun Li, Alexey Kovalev, Chunbai Hu, Xianglong Chen, Lixin Wang, Wenying Li, Yawei Zhou, Hong Zhu Johannes Krause, Alexander Herbig, Yinqiu Cui/ Origin and dispersal history of Hepatitis B virus in Eastern Eurasia // Nat Commun, 2024; 5;15:2951.
- Rute Marcelino, Ifeanyi Jude Ezeonwumelu, André Janeiro, Paula Mimoso, Sónia Matos, Veronica Briz, Victor Pimentel, Marta Pingarilho, Rui Tato Marinho, José Maria Marcelino, Nuno Taveira, Ana Abecasis / . Phylogeography of hepatitis B virus: The role of Portugal in the early dissemination of HBVworldwide // PLoS One. 2022; 17(12):
Acknowledgments
We would like to show our gratitude to the Abbott Transfusion Medicine Medical affairs team for financial support of the publication.
Abstract. Annually 800-900 new cases of hepatitis B chronic forms are registered in Belarus. Compulsory vaccination introduced in the mid-90s certainly produces a positive effect on reducing the number of HBV cases. About 75,000 patients with chronic hepatitis B living in the country are estimated to live in the country at the moment. The main goal of this work was to determine the genotypes/subgenotypes of the hepatitis B virus and establish their role in maintaining the epidemiological process on the territory of Belarus. Serological (CMIA, ELISA), molecular biological (PCR, sequencing) and bioinformatic (phylogenetic analysis) methods have been used. As studies have shown, out of all 884 sequenced and analyzed samples, 722 (81.7%) of them have genotype D; 156 (17.7%) - genotype A; 3 (0.3%) - genotype C and the only 1 (0.1%) – genotype B. For two cases (0.2%) a recombinant form of the hepatitis B virus was detected. The epidemic process of hepatitis B in the country is supported mainly by the circulation of “local” variants of the virus. At the same time, there are occasional introductions of new variants from outside the country.
Keywords. HBV, genotypes/subgenotypes, sequencing, phylogenetic analysis.
Reviewer 3 Report
Comments and Suggestions for Authors
In the introduction, you did not include the classification of the virus, and the genome organization. Please complete this information.
Include the information about the route of HBV transmission.
Please give more date about the HBV vaccination program in Belaruse. When it started? What proportion of the population is vaccinated?
MM - Please include date about the oligonucleotide primers used for the nested-PCR amplification and sequencing of the genomic region of HBV.
Viral RNA/DNA from the blood serum/plasma samples were isolated. What do you mean RNA/DNA? What are you isolating, DNA, RNA, or both? and why
Comments 4: Please provide data on HBV DNA Quantification.
Response 3: We have provided some data on the quantification of HBV DNA in the PCR section.
Your answer is not clear.
What primers did you use for the second run of nested PCR? Please, make a table with your primer pairs and their nucleotide sequences that you used in the first and second round of nested PCR, and reaction conditions.
What size is your fragment on Nested amplification?
882 samples of 884 analysed samples are positive for HBV, which to me is an excessively high success rate for the methods used. What is your explanation?
Author Response
Rev_3
Comments and Suggestions for Authors
Good afternoon,
Dear colleague!
Thank you for your questions. I will try to answer them in more detail.
- In the introduction, you did not include the classification of the virus, and the genome organization. Please complete this information.
HBV belongs to the Hepadnaviridae family, with a genome of approximately 3.2 kb in length.
Now the HBV is divided into ten genotypes (A-J) and 24 subgenotypes (A1–A3, B1–B5, C1–C6, D1–D6 and F1–F4. - A systematic genotype and subgenotype re-ranking of hepatitis B virus under a novel classification standard.
Yin Y, He K, Wu B, Xu M, Du L, Liu W, Liao P, Liu Y, He M.Heliyon. 2019 Oct 23;5(10):e02556
- Include the information about the route of HBV transmission.
Hepatitis B virus, like HIV, has three main transmission mechanisms: parenteral (through blood and its products and/or co-injection of drugs), vertical (from infected mother to child) and sexual transmission.
- Please give more date about the HBV vaccination program in Belarus. When it started? What proportion of the population is vaccinated?
In Belarus, selective hepatitis B vaccination for epidemic indications has been organised since 1996 and included in the national preventive vaccination calendar since 2000. Vaccination of newborns, children aged 13 years and certain risk groups (health care workers and persons in domestic contact with infected persons) has helped to reduce the incidence of hepatitis B infection.
Vaccination against viral hepatitis B has made it possible to reduce the incidence of acute hepatitis B almost six-fold over the past 10 years (from 5.9 to 1.02 per 100,000 population) and to consider Belarus as a country with a low incidence of acute viral hepatitis B (less than 2 per cent of the population).
This means that vaccination coverage against hepatitis B has been over 97 percent for several decades. This is a good result. In 2022, a WHO-supported study of the child population in Belarus showed how many of those vaccinated had HBsAg, the surface antigen of viral hepatitis B: - It was detected in only one child.
- MM - Please include date about the oligonucleotide primers used for the nested-PCR amplification and sequencing of the genomic region of HBV.
Viral RNA/DNA from the blood serum/plasma samples were isolated. What do you mean RNA/DNA? What are you isolating, DNA, RNA, or both? and why
Nucleic acid isolation kits allow both RNA and DNA to be isolated. PCR is then performed using either reverse transcription (for HIV or HCV) or PCR for hepatitis B.
Comments 4: Please provide data on HBV DNA Quantification.
Response 3: We have provided some data on the quantification of HBV DNA in the PCR section.
Your answer is not clear.
Viral load in the samples ranged from 1.2x102 to 3.4x106 IU/ml.
- What primers did you use for the second run of nested PCR? Please, make a table with your primer pairs and their nucleotide sequences that you used in the first and second round of nested PCR, and reaction conditions.
(p1: 5-CCTGCTGGTGGCTCCAGTTC-3 at nucleotide position 55-76, and primer pR5: 5-GGT TGC GTC AGC AAA CAC TTG-3 at position 1197-1178), were used for amplification of the conserved domains Ato E of the viral polymerase gene. A nested PCR was performed on some samples that did not give a successful amplification, using primers p4 5-CTC ACA ATA CCG CAG AGT CTA GAC T-3 at nucleotide position 230-254, and pR2: 5-AAA GCC CAA AAG ACC CAC AAT-3 at position 1017-997.
- What size is your fragment on Nested amplification?
As a result, we obtained a fragment of about 900 base pairs
882? samples of 884 analysed samples are positive for HBV, which to me is an excessively high success rate for the methods used. What is your explanation?
A total of 884 known positive serum/plasma samples were taken. The samples were obtained from patients and blood donors with acute and chronic HBV from different regions of the country.
HEPATITIS B VIRUS GENOTYPES AND SUBGENOTYPES CIRCULATING IN BELARUS
1Vladimir Eremin, 1Fedor Karpenko, 2Igor Karpov, 3Valeriy Semenov, 3Ina Oiestad
1Republican Scientific and Practical Center for Transfusiology and Medical Biotechnology
2Belarusian State Medical University, Department of Infectious Diseases, Belarus
3Vitebsk State Medical University, Department of Infectious Diseases, Belarus
Abstract. Approximately 800-900 new cases of chronic forms of hepatitis B are reported in Belarus annually. Compulsory vaccination, introduced to the country in the mid-90s, produces a certain positive effect on reducing the number of HBV cases. However, around 75,000 patients with chronic hepatitis B are estimated to live in the country at the moment. The main goal of this research was to determine the genotypes/subgenotypes of the hepatitis B virus and establish their role in maintaining the epidemic process in the country. Serological (CMIA, ELISA), molecular biological (PCR, sequencing) and bioinformatic (phylogenetic analysis) methods have been used to obtain results. As studies have shown, in 722 (81.7%) samples genotype D has been determined; in 156 (17.7%) - genotype A; in 3 (0.3%) - genotype C and only in 1 sample (0.1%) – genotype B. In two cases (0.2%) a recombinant form of the hepatitis B virus has been detected. The epidemic process of hepatitis B in the country is supported mainly by the circulation of “local” variants of the virus. At the same time, there are occasional introductions of new genetic variants from the outside of the country.
Keywords. HBV, genotypes/subgenotypes, sequencing, phylogenetic analysis, hepatitis B.
Introduction. Hepatitis B virus (HBV) infection remains a serious health concern despite the HBV vaccine introducing in the late 1990s. Approximately two billion cases of the disease were reported worldwide from the time of HBV discovery to the beginning of the 21st century. More than 400 million of them subsequently became chronic HBV carriers [1]. Chronic HBV infection is considered a major cause of many liver-related medical complications such as hepatocellular carcinoma, liver cirrhosis, and liver failure [2,3,4]. Hepatitis B virus, like HIV, has three main transmission mechanisms: parenteral (through blood and its products and/or co-injection of drugs), vertical (from infected mother to child) and sexual transmission. HBV belongs to the Hepadnaviridae family, with a genome of approximately 3.2 kb in length. Due to the virus complexity, HBV genome contains an overlapping region that encodes four different genes (S, C, P and X). The S gene encodes exclusively the HBV surface envelope protein in its long open reading frame [6]. The presence of start codons allows the gene splicing into small, medium and large polypeptides, which are typically used either as a single target gene or in combination with other HBV genes to detect HBV DNA [5]. HBV core antigen is commonly encoded by the C gene, whereas the X gene encodes protein X [7]. The P gene encodes a polymerase protein, an integral component of the reverse transcription process during HBV replication. The HBV replication cycle lacks proofreading properties, resulting in progeny with a high genomic variability [5].
Due to significant genetic diversity, HBV is categorized into ten genotypes (A-J) with 7.5 % intergroup variation [8]. In addition to E and G, all genotypes are classified into 25 subgenotypes with 4% amino acid variability [9,10]. HBV genotypes are distributed differently depending on the geography: HBV-B, HBV-C and HBV-E are most common in Oceania and East Asia, while HBV-E is most prevalent in Central and West Africa. HBV-F and HBV H are found only in Alaska and Latin America. In contrast, HBV-D is a global pandemic. In Australia, Europe, Indonesia, North Africa and Western Asia, HBV-D1 is the most common virus, while HBV-D2 is found in Albania, Japan, Malaysia, North-Eastern Europe, Russia and the UK [11,12,13,14,15]. Now the HBV is divided into ten genotypes (A-J) and 24 subgenotypes (A1–A3, B1–B5, C1–C6, D1–D6 and F1–F4. A systematic genotype and subgenotype re-ranking of hepatitis B virus under a novel classification standard [16].
The progression and natural course of the disease varies among different HBV genotypes. These factors can complicate HBV infection treatment since the efficacy of known therapeutics turns out to be ineffective against certain genotypes and new genotypic variants [8,9]. Thus, there is a global exigency for constant update in genotypic information and surveys of HBV-infected patients [15].
In Belarus, selective hepatitis B vaccination for epidemic indications has been organised since 1996 and included in the national preventive vaccination calendar since 2000. Vaccination of newborns, children aged 13 years and certain risk groups (health care workers and persons in domestic contact with infected persons) has helped to reduce the incidence of hepatitis B infection. Vaccination against viral hepatitis B has made it possible to reduce the incidence of acute hepatitis B almost six-fold over the past 10 years (from 5.9 to 1.02 per 100,000 population) and to consider Belarus as a country with a low incidence of acute viral hepatitis B (less than 2 per cent of the population). This means that vaccination coverage against hepatitis B has been over 97 percent for several decades. This is a good result. In 2022, a WHO-supported study of the child population in Belarus showed how many of those vaccinated had HBsAg, the surface antigen of viral hepatitis B: - It was detected in only one child. The most affected age groups are teenagers and adults 15 to 40 years old. The incidence rate in this specific age group is significantly higher than in other age groups of the population of the Republic. Approximately 800-900 new cases of chronic viral hepatitis B are detected annually in the country, despite a significant decrease of occurrence of acute viral hepatitis B. According to the epidemiological essays, 75,000 patients with chronic hepatitis B currently live in Belarus.
Materials and methods Samples. 884 blood serum/plasma samples were tested for HBV markers with CMIA /ELISA and PCR methods. The samples were obtained from patients and blood donors with acute and chronic HBV from different regions of the country. The age of the patients ranged from 4 years to 78 years. 404 samples were obtained from males and 250 – from females. In 230 cases gender and age were not specified. Serum/plasma samples were collected and analyzed during 2014-2023. Serological tests. The commercial kit chemiluminescent microparticle immunoassay (CMIA) Architect HBsAg Qualitative II Reagent Kit, Abbott, USA was applied to detect HBsAg. Positive results were confirmed using the Architect HBsAg Qualitative II Confirmatory Reagent Kit, Abbott, USA as well as ELISA “Vectogep B-HBs antigen», manufactured by “Vector-Best”, Novosibirsk, Russia.
PCR. The polymerase chain reaction was used to determine DNA of the hepatitis B virus using test systems, produced by Vector-Best CJSC: “RealBest HBV DNA (quantitative”, “RealBest HIV RNA (quantitative”) in accordance with the manufacturer’s instructions. Viral load in the samples ranged from 1.2x102 to 3.4x106 IU/ml HBV DNA.
Primers. We used the previously described by L Serfaty et al [17] primer pairs for sequencing the S and P regions of the hepatitis B virus genome. DNA isolation. Viral RNA/DNA from the blood serum/plasma samples were isolated using the “Kit of reagents for the isolation of NK” (manufactured by “Vector-Best”, Russia) and the “Kit of reagents for the isolation of RNA/DNA from clinical material” “RIBO- sorb", "RNA-prep", Russia. The manufacturers’ instructions were strictly followed. Nested PCR was performed using previously described primer pairs p1/pR5 (p1: 5-CCTGCTGGTGGCTCCAGTTC-3 at nucleotide position 55-76, and primer pR5: 5-GGT TGC GTC AGC AAA CAC TTG-3 at position 1197-1178) and p4/pR2 (p4 5-CTC ACA ATA CCG CAG AGT CTA GAC T-3 at nucleotide position 230-254, and pR2: 5-AAA GCC CAA AAG ACC CAC AAT-3 at position 1017-997) [17] according to the following protocol in a volume of 25µl: 2,5 µl 10х buffer+ MgCl2; 0,25 µl 25mM dNTPs; 0,5 µl 10 µM p1; 0,5 µl 10 µM pR5; 0,8 10 µM Taq polymerase; 18,45 µl bdH20. The first round of PCR was carried out according to the following protocol: one denaturation cycle at 95oC for 2 minutes and 30 cycles of amplification were performed with denaturation at 95°C for 30 sec, annealing at 53°C for 30 sec, and extension at 68°C for 1 minutes, then one cycle at 68oC for 5 minutes and storage at 10oC. In the second round of PCR, the annealing temperature was 50°C; the rest parameters were not changed. As a result, we obtained a fragment of about 900 base pairs Amplified DNA fragments were analyzed on a 2% agarose gel. Electrophoresis was carried out at 10 w/cm of gel in TRIS-acetate buffer, pH 8.0. DNA was visualized using the Vitran Photo gel documentation system (Biocom Company LLC, Russia). The fragment size was determined according to the molecular weight of marker of 100-1000 bp (Fermentas, Lithuania).
The resulting DNA fragments were purified using the NimaGen ExS-Pure kit, Holland.
PCR sequencing was carried out using the second round primer pair p4/pR2 in the volume of 10 µl according to the following protocol: 5x seqbuf - 2 µl; BigDye Terminator v. 3.1 (applied biosystems) - 1 µl; p4/pR4 - 2 µl; bdH20 - 3 bdH20; DNA - 2 µl.
Electrophoresis of HBV DNA fragments obtained and purified after PCR sequencing was performed on AB 3500 genetic analyzer, USA. Phylogenetic analysis of the obtained HBV DNA fragments was conducted using the computer programs: Sequencing Analysis v.6, BioEdit, SeqScape v3, MEGAX and Genious 8.1. Phylogenetic trees were built using the ML (maximum likelihood) algorithm in the PHYML program. The SH-aLRT test was performed to calculate the statistical significance of the clusters. Clusters with a support node ≥0.9 were considered reliable. Mutations in the HBV genome in the S and P regions were determined using the programs: https://www.geno2pheno.org, http://www.hiv-grade.de/hbv_grade and https://hivdb.stanford.edu/HBV/HBVseq. Results The studies showed the following: 722 (81.7%) out of all 884 sequenced and analyzed samples, had genotype D; 156 (17.7%) - genotype A; 3 (0.3%) - genotype C and the only 1 (0.1%) – genotype B. For two cases (0.2%) a recombinant form of the hepatitis B virus was detected. Genotype D was found to be represented by subgenotypes - D1 (145/20.1%), D2 (371/51.4%), D3 (201/27, 8%) and D4 (5/0.7%); A – A2; C – C1 and C2; B – B4; recombinant forms C2/ D and A2/ С2 were detected (Figure 1). Figure 1. HBV subgenotypes identified in Belarus In 12 cases, the virus with mutation mutations to reverse transcriptase inhibitors Lamivudine, Zeffix® and Telbivudine, Tyzeka®, Sebivo® and partial resistance to Entecavir, Baraclude® was identified. Moreover, in 4 cases, resistance of mutations was identified in subgenotypes A2 and D2, in 3 cases- in the genome of subgenotype D1, and in one case - in subgenotype D3. Mutations were most often recorded at positions 180M and 204V - in 8 cases, 204I - in three cases, 80I and 173L - 2 times each, and 80V and 181T in one case (Table 1). Table 1 – Resistance of mutations identified in different HBV subgenotypes
|
N |
Subgenotypes |
Mutations |
|
1. |
А2 |
80V-1; 180M-3; 181T-1; 204V-3; |
|
2. |
D1 |
173L-1; 180M-3; 204V-3; |
|
3. |
D2 |
80I-2; 173L-1; 180M-2; 204V-2; 204I-2; |
|
4. |
D3 |
204I-1; |
Determining mutations is very important for infectious disease doctors, since substitutions at positions 180M, 181T, 204V and 204I lead to resistance to Lamivudine (Zeffix), Adefovir (Hepsera) and Telbivudine (Tyzeka, Sebivo) and require a change in treatment regimen.
Phylogenetic analysis of subtype A2 DNA sequences showed that most samples from Belarus were found nearby and formed at least 7 clusters (Figures 2, 3, 4). At the same time, some samples were in different groups and formed clusters with samples from the USA, Cuba, Brazil, similarly to Western European countries, mainly from Poland, Italy, France, Belgium, Germany and some others (Figures 2,3,4).
|
2 |
|
3 |
|
4 |
Fig.2, 3, 4 Phylogenetic analysis of subtype А2 (Here and in other figures, samples from Belarus are indicated in red)
12 clusters were identified while analyzing samples of subtype D1. Most of the sequenced and analyzed HBV DNA samples were identified with sequences from Pakistan, India, Iran, Bangladesh and some European countries such as France, Holland, Italy. Two large clusters, consisting mainly of samples from Belarus, were found with sequences from Iran and Italy, as well as Pakistan and France, Oman, Holland and Russia (Figures 5,6).
|
5 |
|
6 |
Fig. 5 and 6 Phylogenetic analysis of subtype D1
Sequences of subtype D2 formed 14 clusters, which were present in the samples from Belarus described earlier. They showed similarities with samples from Estonia, USA, India, Spain, Russia, Cameroon, Latvia, Iran and Belgium, Figure 7, 8, 9.
|
7 |
|
8 |
|
9 |
Fig. 7, 8, 9 Phylogenetic analysis of subtype D2
Phylogenetic analysis of D3 subtype showed that the samples formed 14 clusters mainly originated from India, Croatia, Russia, Italy, Germany and even Brazil. The largest cluster included samples from Belarus, with the previously described sequences. These clusters originated from Poland, Russia, Rwanda, Estonia and Albania, Figure 10.
Fig. 10 Phylogenetic analysis of subtype D3
The phylogenetic analysis of the subtype D4 samples showed that they all were found in the same group and clustered with HBV sequences from Cuba and Haiti, Figure 11.
Fig. 11 Phylogenetic analysis of subtype D4
Samples of subgenotypes C1 and C2 were identified in adult patients and a child from China. Sequences of subtype C1 formed a cluster, with samples from the USA and Vietnam, and subtype C2 with reference samples from South Korea, USA and China, Figure 12,13.
Fig .12, 13. Phylogenetic analysis of subtypes C1 and C2
The subtype B4 sample was revealed in a student from Vietnam who came to study in our country.
Finally, both samples with recombinant forms of HBV were identified in residents of Belarus
Discussion
Approximately 800-900 new cases of chronic forms of hepatitis B are reported in Belarus annually. Compulsory vaccination, introduced to the country in the mid-90s, produces a certain positive effect on reducing the number of HBV cases. However, around 75,000 patients with chronic hepatitis B are estimated to live in the country at the moment. Our previous study showed that subgenotype D2 (55.8%) was dominant in Belarus, followed by D3 (18,3%), D1 (11,6%) and A2 (11,6%) [18]. The conducted studies show that the structure of genotypes/subgenotypes in the country has actually remained unchanged. Genotype D still dominates in the country and accounts for 81.7% of all analyzed cases. Genotype D was found to be represented by subgenotypes D1 (145/20.1%), D2 (371/51.4%), D3 (201/27, 8%) and D4 (5/0.7%); A – A2; C – C2; B – B4; recombinant forms C2/ D and A2/ С2 were detected. Possible causes of the infection in patients were determined i.e. they occurred due to the circulation of “domestic” variants of the virus and/or the introduction of the virus from outside. The findings obtained show that the epidemic transmission of viral hepatitis B in the country is maintained mainly due to the circulation of local variants of the virus. It is confirmed by the examples of subgenotypes A2, D1, D2 and D3, where the largest clusters were formed by samples from Belarus. However, it should be noted that new variants of HBV are constantly being introduced into the country. The main routes of introduction of A2 subgenotype are Western Europe countries and even the USA, Cuba, Brazil; for D1 - Pakistan, India, Iran, Bangladesh, Italy, Holland and Russia; D2 - Estonia, USA, India, Spain, Russia, Cameroon, Latvia, Iran and Belgium; D3 - Poland, Russia, Rwanda, Estonia and Albania. All samples of D4 subgenotype were clustered with sequences from Cuba and Haiti, C1 and C2 – from China, and B4 – from Vietnam. Two recombinant forms C2/D and A2/D were identified in the residents of our country. Comparing the distribution of different HBV genotypes in neighboring countries and even in different regions of the same country, we may conclude that they vary. For example, genotype D significantly dominates in the European part of Russian Federation, while, for example, in Yakutia, genotype A occupies one of the leading positions (36.4% - A and 58,6% - D) [19]. In Asia, particularly in its eastern part, genotypes B and C dominate, and the C2 subgenotype is epidemic in China [20, 21]. In Western Europe, in particular in Portugal HBV genotype A (HBV/A) was the most prevalent genotype (41.5%), followed by D [HBV/D; (33.8%)], and E [HBV/E; (24.6%)]. Subgenotypes A1 (HBV/A1) and D4 (HBV/D4) were almost equally prevalent with 23.1% and 22.3%, respectively, followed by A2 (HBV/A2) with 16.2% and D3 (HBV/D3) with 11.5% [22].
The Republic of Belarus is geographically located in the center of Europe. The residents of the country are actively travelling and many people from other countries relocate to Belarus for work and study. This fact can easily explain the introduction of the new variants of HBV to the Republic.
The results of this study contribute valuable information of the molecular genetic characteristics of HBV, prevailing in the Republic of Belarus. Possible routes of the introduction of the new variants of HBV have been identified. The main mutations in the P region of the virus genome leading to resistance to antiviral drugs have detected. Identification of the mutations at positions 181T, 204V, 204I, which determine resistance not only to Lamivudine (Zefffix), but also to Adefovir (Hepsera) and Telbivudine (Tyzeka, Sebivo), made it possible to use Tenofovir DF for treatment of the patients. The compensatory mutations 80V, 173L, 80I have been found in the complex with the mutations leading to virus resistance to antiviral drugs.
Conclusions
To summarize, should be noted that the use of the methods of molecular epidemiology solve many issues for both: epidemiologists and infectious diseases specialists. This knowledge is essential to plan preventive measures for the directions and routes of its’ spread in the country by determining the genotypes/subgenotypes of the hepatitis B virus. The quality of living of our patients could be improved by prescribing new treatment regimens, based on the resistance mutations identification.
REFERENCES
- Thun M.J., DeLancey J.O., Center M.M., Jemal A., Ward E.M. The global burden of cancer: Priorities for prevention. Carcinogenesis. 2010;31:100–110.
- Philips C.A., Ahamed R., Abduljaleel J.K., Rajesh S., Augustine P. Critical Updates on Chronic Hepatitis B Virus Infection in 2021. Cureus. 2021;13:e19152.
- Kao J.-H., Chen P.-J., Lai M.-Y., Chen D.-S. Hepatitis B genotypes correlate with clinical outcomes in patients with chronic hepatitis B. Gastroenterology. 2000;118:554–559.
- Aghakhani A., Hamkar R., Zamani N., Eslamifar A., Banifazl M., Saadat A., Sofian M., Adibi L., Irani N., Mehryar M. Hepatitis B virus genotype in Iranian patients with hepatocellular carcinoma. Int. J. Infect. Dis. 2009;13:685–689.
- Beck J., Nassal M. Hepatitis B virus replication. World J. Gastroenterol. 2007;13:48–64.
- Pollicino T., Cacciola I., Saffioti F., Raimondo G. Hepatitis B virus PreS/S gene variants: Pathobiology and clinical implications. J. Hepatol. 2014;61:408–417.
- Ie S.I., Thedja M.D., Roni M., Muljono D.H. Prediction of conformational changes by single mutation in the hepatitis B virus surface antigen (HBsAg) identified in HBsAg-negative blood donors. Virol. J. 2010;7:326.
- Lin C.-L., Kao J.-H. The clinical implications of hepatitis B virus genotype: Recent advances. J. Gastroenterol. Hepatol. 2011;26((Suppl. 1)):123–130.
- Zhu S.S., Zhang H.F., Wang H.B., Dong Y., Chen D., Jia W., Gan Y., Chen J. Relation of viral genotypes to clinical features in children with chronic hepatitis B. Chin. J. Exp. Clin. Virol. 2008;22:192–194.
- Shamseer L., Moher D., Clarke M., Ghersi D., Liberati A., Petticrew M., Shekelle P., Stewart L.A. Preferred reporting items for systematic review and meta-analysis protocols (PRISMA-P) 2015: Elaboration and explanation. BMJ. 2015;349:g7647.
- George B.J., Aban I.B. An application of meta-analysis based on DerSimonian and Laird method. J. Nucl. Cardiol. 2016;23:690–692.
- Fletcher J. What is heterogeneity and is it important? BMJ. 2007;334:94–96.
- Higgins J.P.T., Thompson S.G., Deeks J.J., Altman D.G. Measuring inconsistency in meta-analyses. BMJ. 2003;327:557–560.
- Wallace B.C., Dahabreh I.J., Trikalinos T.A., Lau J., Trow P., Schmid C.H. Closing the Gap between Methodologists and End-Users: R as a Computational Back-End. J. Stat. Softw. 2012;49:2–15.
- Munn Z., MClinSc S.M., Lisy K., Riitano D., Tufanaru C. Methodological guidance for systematic reviews of observational epidemiological studies reporting prevalence and cumulative incidence data. Int. J. Evid. Based Healthc. 2015;13:147–153.
- Yin Y, He K, Wu B, Xu M, Du L, Liu W, Liao P, Liu Y, He M. /Heliyon - 2019 Oct 23;5(10):e02556.
- Serfaty L., Thabut D., Zoulim F., Andreani T., Eres O. C., Carbonell N., Loria A., and Poupon R. / Hepatology – 2001 – V.34 – N.- 3 – P.573-577.
- Olinger C.M., Lazouskaya N.V., Eremin V.F., Muller C.P. Multiple genotypes of hepatitis B and C viruses in Belarus: similarities with Russia and European influences. Clin Microbiol Infect – 2008; 14:575-581.
- Anastasia A Karlsen, Karen K Kyuregyan, Olga V Isaeva, Vera S Kichatova, Fedor A Asadi Mobarkhan, Lyudmila V Bezuglova, Irina G Netesova, Victor A Manuylov, Andrey A Pochtovyi, Vladimir A Gushchin, Snezhana S Sleptsova, Margarita E Ignateva, Mikhail I Mikhailov Different evolutionary dynamics of hepatitis B virus genotypes A and D, and hepatitis D virus genotypes 1 and 2 in an endemic area of Yakutia, Russia // /BMC Infect Dis – 2022; 12;22:452
- Kizito Eneye Bello, Tuan Nur Akmalina Mat Jusoh, Ahmad Adebayo Irekeola, Norhidayah Abu, Nur Amalin Zahirah Mohd Amin, Nazri Mustaffa, Rafidah Hanim Shueb/ A Recent Prevalence of Hepatitis B Virus (HBV) Genotypes and Subtypes in Asia: A Systematic Review and Meta-Analysis // Healthcare (Basel) – 2023; 1;11(7):1011.
- Bing Sun, Aida Andrades Valtueña, Arthur Kocher, Shizhu Gao, Chunxiang Li, Shuang Fu, Fan Zhang, Pengcheng Ma, Xuan Yang, Yulan Qiu, Quanchao Zhang, Jian Ma, Shan Chen, Xiaoming Xiao, Sodnomjamts Damchaabadgar, Fajun Li, Alexey Kovalev, Chunbai Hu, Xianglong Chen, Lixin Wang, Wenying Li, Yawei Zhou, Hong Zhu Johannes Krause, Alexander Herbig, Yinqiu Cui/ Origin and dispersal history of Hepatitis B virus in Eastern Eurasia // Nat Commun, 2024; 5;15:2951.
- Rute Marcelino, Ifeanyi Jude Ezeonwumelu, André Janeiro, Paula Mimoso, Sónia Matos, Veronica Briz, Victor Pimentel, Marta Pingarilho, Rui Tato Marinho, José Maria Marcelino, Nuno Taveira, Ana Abecasis / . Phylogeography of hepatitis B virus: The role of Portugal in the early dissemination of HBVworldwide // PLoS One. 2022; 17(12):
Acknowledgments
We would like to show our gratitude to the Abbott Transfusion Medicine Medical affairs team for financial support of the publication.
Abstract. Annually 800-900 new cases of hepatitis B chronic forms are registered in Belarus. Compulsory vaccination introduced in the mid-90s certainly produces a positive effect on reducing the number of HBV cases. About 75,000 patients with chronic hepatitis B living in the country are estimated to live in the country at the moment. The main goal of this work was to determine the genotypes/subgenotypes of the hepatitis B virus and establish their role in maintaining the epidemiological process on the territory of Belarus. Serological (CMIA, ELISA), molecular biological (PCR, sequencing) and bioinformatic (phylogenetic analysis) methods have been used. As studies have shown, out of all 884 sequenced and analyzed samples, 722 (81.7%) of them have genotype D; 156 (17.7%) - genotype A; 3 (0.3%) - genotype C and the only 1 (0.1%) – genotype B. For two cases (0.2%) a recombinant form of the hepatitis B virus was detected. The epidemic process of hepatitis B in the country is supported mainly by the circulation of “local” variants of the virus. At the same time, there are occasional introductions of new variants from outside the country.
Keywords. HBV, genotypes/subgenotypes, sequencing, phylogenetic analysis.
Round 3
Reviewer 1 Report
Comments and Suggestions for Authors
No comments.
Author Response
The reviewer has no comment
Reviewer 3 Report
Comments and Suggestions for Authors
Please include the references L33-35
Please include the references L36-37
Please include the references L66-69
The quality of figures 2-9 is inferior, and new figures of higher quality need to be made.
Author Response
Rev_3
Comments and Suggestions for Authors
Good afternoon,
Dear colleague!
Thank you for your questions. I will try to answer them in more detail.
- In the introduction, you did not include the classification of the virus, and the genome organization. Please complete this information.
HBV belongs to the Hepadnaviridae family, with a genome of approximately 3.2 kb in length.
Now the HBV is divided into ten genotypes (A-J) and 24 subgenotypes (A1–A3, B1–B5, C1–C6, D1–D6 and F1–F4. - A systematic genotype and subgenotype re-ranking of hepatitis B virus under a novel classification standard.
Yin Y, He K, Wu B, Xu M, Du L, Liu W, Liao P, Liu Y, He M.Heliyon. 2019 Oct 23;5(10):e02556
- Include the information about the route of HBV transmission.
Hepatitis B virus, like HIV, has three main transmission mechanisms: parenteral (through blood and its products and/or co-injection of drugs), vertical (from infected mother to child) and sexual transmission.
- Please give more date about the HBV vaccination program in Belarus. When it started? What proportion of the population is vaccinated?
In Belarus, selective hepatitis B vaccination for epidemic indications has been organised since 1996 and included in the national preventive vaccination calendar since 2000. Vaccination of newborns, children aged 13 years and certain risk groups (health care workers and persons in domestic contact with infected persons) has helped to reduce the incidence of hepatitis B infection.
Vaccination against viral hepatitis B has made it possible to reduce the incidence of acute hepatitis B almost six-fold over the past 10 years (from 5.9 to 1.02 per 100,000 population) and to consider Belarus as a country with a low incidence of acute viral hepatitis B (less than 2 per cent of the population).
This means that vaccination coverage against hepatitis B has been over 97 percent for several decades. This is a good result. In 2022, a WHO-supported study of the child population in Belarus showed how many of those vaccinated had HBsAg, the surface antigen of viral hepatitis B: - It was detected in only one child.
HEPATITIS B VIRUS GENOTYPES AND SUBGENOTYPES CIRCULATING IN BELARUS
1Vladimir Eremin, 1Fedor Karpenko, 2Igor Karpov, 3Valeriy Semenov, 3Ina Oiestad
1Republican Scientific and Practical Center for Transfusiology and Medical Biotechnology
2Belarusian State Medical University, Department of Infectious Diseases, Belarus
3Vitebsk State Medical University, Department of Infectious Diseases, Belarus
Abstract. Approximately 800-900 new cases of chronic forms of hepatitis B are reported in Belarus annually. Compulsory vaccination, introduced to the country in the mid-90s, produces a certain positive effect on reducing the number of HBV cases. However, around 75,000 patients with chronic hepatitis B are estimated to live in the country at the moment. The main goal of this research was to determine the genotypes/subgenotypes of the hepatitis B virus and establish their role in maintaining the epidemic process in the country. Serological (CMIA, ELISA), molecular biological (PCR, sequencing) and bioinformatic (phylogenetic analysis) methods have been used to obtain results. As studies have shown, in 722 (81.7%) samples genotype D has been determined; in 156 (17.7%) - genotype A; in 3 (0.3%) - genotype C and only in 1 sample (0.1%) – genotype B. In two cases (0.2%) a recombinant form of the hepatitis B virus has been detected. The epidemic process of hepatitis B in the country is supported mainly by the circulation of “local” variants of the virus. At the same time, there are occasional introductions of new genetic variants from the outside of the country.
Keywords. HBV, genotypes/subgenotypes, sequencing, phylogenetic analysis, hepatitis B.
Introduction. Hepatitis B virus (HBV) infection remains a serious health concern despite the HBV vaccine introducing in the late 1990s. Approximately two billion cases of the disease were reported worldwide from the time of HBV discovery to the beginning of the 21st century. More than 400 million of them subsequently became chronic HBV carriers [1]. Chronic HBV infection is considered a major cause of many liver-related medical complications such as hepatocellular carcinoma, liver cirrhosis, and liver failure [2,3,4]. Hepatitis B virus, like HIV, has three main transmission mechanisms: parenteral (through blood and its products and/or co-injection of drugs), vertical (from infected mother to child) and sexual transmission. HBV belongs to the Hepadnaviridae family, with a genome of approximately 3.2 kb in length. Due to the virus complexity, HBV genome contains an overlapping region that encodes four different genes (S, C, P and X). The S gene encodes exclusively the HBV surface envelope protein in its long open reading frame [6]. The presence of start codons allows the gene splicing into small, medium and large polypeptides, which are typically used either as a single target gene or in combination with other HBV genes to detect HBV DNA [5]. HBV core antigen is commonly encoded by the C gene, whereas the X gene encodes protein X [7]. The P gene encodes a polymerase protein, an integral component of the reverse transcription process during HBV replication. The HBV replication cycle lacks proofreading properties, resulting in progeny with a high genomic variability [5].
Due to significant genetic diversity, HBV is categorized into ten genotypes (A-J) with 7.5 % intergroup variation [8]. In addition to E and G, all genotypes are classified into 25 subgenotypes with 4% amino acid variability [9,10]. HBV genotypes are distributed differently depending on the geography: HBV-B, HBV-C and HBV-E are most common in Oceania and East Asia, while HBV-E is most prevalent in Central and West Africa. HBV-F and HBV H are found only in Alaska and Latin America. In contrast, HBV-D is a global pandemic. In Australia, Europe, Indonesia, North Africa and Western Asia, HBV-D1 is the most common virus, while HBV-D2 is found in Albania, Japan, Malaysia, North-Eastern Europe, Russia and the UK [11,12,13,14,15]. Now the HBV is divided into ten genotypes (A-J) and 24 subgenotypes (A1–A3, B1–B5, C1–C6, D1–D6 and F1–F4. A systematic genotype and subgenotype re-ranking of hepatitis B virus under a novel classification standard [16].
The progression and natural course of the disease varies among different HBV genotypes. These factors can complicate HBV infection treatment since the efficacy of known therapeutics turns out to be ineffective against certain genotypes and new genotypic variants [8,9]. Thus, there is a global exigency for constant update in genotypic information and surveys of HBV-infected patients [15].
In Belarus, selective hepatitis B vaccination for epidemic indications has been organised since 1996 and included in the national preventive vaccination calendar since 2000. Vaccination of newborns, children aged 13 years and certain risk groups (health care workers and persons in domestic contact with infected persons) has helped to reduce the incidence of hepatitis B infection. Vaccination against viral hepatitis B has made it possible to reduce the incidence of acute hepatitis B almost six-fold over the past 10 years (from 5.9 to 1.02 per 100,000 population) and to consider Belarus as a country with a low incidence of acute viral hepatitis B (less than 2 per cent of the population). This means that vaccination coverage against hepatitis B has been over 97 percent for several decades. This is a good result. In 2022, a WHO-supported study of the child population in Belarus showed how many of those vaccinated had HBsAg, the surface antigen of viral hepatitis B: - It was detected in only one child. The most affected age groups are teenagers and adults 15 to 40 years old. The incidence rate in this specific age group is significantly higher than in other age groups of the population of the Republic. Approximately 800-900 new cases of chronic viral hepatitis B are detected annually in the country, despite a significant decrease of occurrence of acute viral hepatitis B. According to the epidemiological essays, 75,000 patients with chronic hepatitis B currently live in Belarus.
Materials and methods Samples. 884 blood serum/plasma samples were tested for HBV markers with CMIA /ELISA and PCR methods. The samples were obtained from patients and blood donors with acute and chronic HBV from different regions of the country. The age of the patients ranged from 4 years to 78 years. 404 samples were obtained from males and 250 – from females. In 230 cases gender and age were not specified. Serum/plasma samples were collected and analyzed during 2014-2023. Serological tests. The commercial kit chemiluminescent microparticle immunoassay (CMIA) Architect HBsAg Qualitative II Reagent Kit, Abbott, USA was applied to detect HBsAg. Positive results were confirmed using the Architect HBsAg Qualitative II Confirmatory Reagent Kit, Abbott, USA as well as ELISA “Vectogep B-HBs antigen», manufactured by “Vector-Best”, Novosibirsk, Russia.
PCR. The polymerase chain reaction was used to determine DNA of the hepatitis B virus using test systems, produced by Vector-Best CJSC: “RealBest HBV DNA (quantitative”, “RealBest HIV RNA (quantitative”) in accordance with the manufacturer’s instructions. Viral load in the samples ranged from 1.2x102 to 3.4x106 IU/ml HBV DNA.
Primers. We used the previously described by L Serfaty et al [17] primer pairs for sequencing the S and P regions of the hepatitis B virus genome. DNA isolation. Viral RNA/DNA from the blood serum/plasma samples were isolated using the “Kit of reagents for the isolation of NK” (manufactured by “Vector-Best”, Russia) and the “Kit of reagents for the isolation of RNA/DNA from clinical material” “RIBO- sorb", "RNA-prep", Russia. The manufacturers’ instructions were strictly followed. Nested PCR was performed using previously described primer pairs p1/pR5 (p1: 5-CCTGCTGGTGGCTCCAGTTC-3 at nucleotide position 55-76, and primer pR5: 5-GGT TGC GTC AGC AAA CAC TTG-3 at position 1197-1178) and p4/pR2 (p4 5-CTC ACA ATA CCG CAG AGT CTA GAC T-3 at nucleotide position 230-254, and pR2: 5-AAA GCC CAA AAG ACC CAC AAT-3 at position 1017-997) [17] according to the following protocol in a volume of 25µl: 2,5 µl 10х buffer+ MgCl2; 0,25 µl 25mM dNTPs; 0,5 µl 10 µM p1; 0,5 µl 10 µM pR5; 0,8 10 µM Taq polymerase; 18,45 µl bdH20. The first round of PCR was carried out according to the following protocol: one denaturation cycle at 95oC for 2 minutes and 30 cycles of amplification were performed with denaturation at 95°C for 30 sec, annealing at 53°C for 30 sec, and extension at 68°C for 1 minutes, then one cycle at 68oC for 5 minutes and storage at 10oC. In the second round of PCR, the annealing temperature was 50°C; the rest parameters were not changed. As a result, we obtained a fragment of about 900 base pairs Amplified DNA fragments were analyzed on a 2% agarose gel. Electrophoresis was carried out at 10 w/cm of gel in TRIS-acetate buffer, pH 8.0. DNA was visualized using the Vitran Photo gel documentation system (Biocom Company LLC, Russia). The fragment size was determined according to the molecular weight of marker of 100-1000 bp (Fermentas, Lithuania).
The resulting DNA fragments were purified using the NimaGen ExS-Pure kit, Holland.
PCR sequencing was carried out using the second round primer pair p4/pR2 in the volume of 10 µl according to the following protocol: 5x seqbuf - 2 µl; BigDye Terminator v. 3.1 (applied biosystems) - 1 µl; p4/pR4 - 2 µl; bdH20 - 3 bdH20; DNA - 2 µl.
Electrophoresis of HBV DNA fragments obtained and purified after PCR sequencing was performed on AB 3500 genetic analyzer, USA. Phylogenetic analysis of the obtained HBV DNA fragments was conducted using the computer programs: Sequencing Analysis v.6, BioEdit, SeqScape v3, MEGAX and Genious 8.1. Phylogenetic trees were built using the ML (maximum likelihood) algorithm in the PHYML program. The SH-aLRT test was performed to calculate the statistical significance of the clusters. Clusters with a support node ≥0.9 were considered reliable. Mutations in the HBV genome in the S and P regions were determined using the programs: https://www.geno2pheno.org, http://www.hiv-grade.de/hbv_grade and https://hivdb.stanford.edu/HBV/HBVseq. Results The studies showed the following: 722 (81.7%) out of all 884 sequenced and analyzed samples, had genotype D; 156 (17.7%) - genotype A; 3 (0.3%) - genotype C and the only 1 (0.1%) – genotype B. For two cases (0.2%) a recombinant form of the hepatitis B virus was detected. Genotype D was found to be represented by subgenotypes - D1 (145/20.1%), D2 (371/51.4%), D3 (201/27, 8%) and D4 (5/0.7%); A – A2; C – C1 and C2; B – B4; recombinant forms C2/ D and A2/ С2 were detected (Figure 1). Figure 1. HBV subgenotypes identified in Belarus In 12 cases, the virus with mutation mutations to reverse transcriptase inhibitors Lamivudine, Zeffix® and Telbivudine, Tyzeka®, Sebivo® and partial resistance to Entecavir, Baraclude® was identified. Moreover, in 4 cases, resistance of mutations was identified in subgenotypes A2 and D2, in 3 cases- in the genome of subgenotype D1, and in one case - in subgenotype D3. Mutations were most often recorded at positions 180M and 204V - in 8 cases, 204I - in three cases, 80I and 173L - 2 times each, and 80V and 181T in one case (Table 1). Table 1 – Resistance of mutations identified in different HBV subgenotypes
|
N |
Subgenotypes |
Mutations |
|
1. |
А2 |
80V-1; 180M-3; 181T-1; 204V-3; |
|
2. |
D1 |
173L-1; 180M-3; 204V-3; |
|
3. |
D2 |
80I-2; 173L-1; 180M-2; 204V-2; 204I-2; |
|
4. |
D3 |
204I-1; |
Determining mutations is very important for infectious disease doctors, since substitutions at positions 180M, 181T, 204V and 204I lead to resistance to Lamivudine (Zeffix), Adefovir (Hepsera) and Telbivudine (Tyzeka, Sebivo) and require a change in treatment regimen.
Phylogenetic analysis of subtype A2 DNA sequences showed that most samples from Belarus were found nearby and formed at least 7 clusters (Figures 2, 3, 4). At the same time, some samples were in different groups and formed clusters with samples from the USA, Cuba, Brazil, similarly to Western European countries, mainly from Poland, Italy, France, Belgium, Germany and some others (Figures 2,3,4).
|
2 |
|
3 |
|
4 |
Fig.2, 3, 4 Phylogenetic analysis of subtype А2 (Here and in other figures, samples from Belarus are indicated in red)
12 clusters were identified while analyzing samples of subtype D1. Most of the sequenced and analyzed HBV DNA samples were identified with sequences from Pakistan, India, Iran, Bangladesh and some European countries such as France, Holland, Italy. Two large clusters, consisting mainly of samples from Belarus, were found with sequences from Iran and Italy, as well as Pakistan and France, Oman, Holland and Russia (Figures 5,6).
|
5 |
|
6 |
Fig. 5 and 6 Phylogenetic analysis of subtype D1
Sequences of subtype D2 formed 14 clusters, which were present in the samples from Belarus described earlier. They showed similarities with samples from Estonia, USA, India, Spain, Russia, Cameroon, Latvia, Iran and Belgium, Figure 7, 8, 9.
|
7 |
|
8 |
|
9 |
Fig. 7, 8, 9 Phylogenetic analysis of subtype D2
Phylogenetic analysis of D3 subtype showed that the samples formed 14 clusters mainly originated from India, Croatia, Russia, Italy, Germany and even Brazil. The largest cluster included samples from Belarus, with the previously described sequences. These clusters originated from Poland, Russia, Rwanda, Estonia and Albania, Figure 10.
Fig. 10 Phylogenetic analysis of subtype D3
The phylogenetic analysis of the subtype D4 samples showed that they all were found in the same group and clustered with HBV sequences from Cuba and Haiti, Figure 11.
Fig. 11 Phylogenetic analysis of subtype D4
Samples of subgenotypes C1 and C2 were identified in adult patients and a child from China. Sequences of subtype C1 formed a cluster, with samples from the USA and Vietnam, and subtype C2 with reference samples from South Korea, USA and China, Figure 12,13.
Fig .12, 13. Phylogenetic analysis of subtypes C1 and C2
The subtype B4 sample was revealed in a student from Vietnam who came to study in our country.
Finally, both samples with recombinant forms of HBV were identified in residents of Belarus
Discussion
Approximately 800-900 new cases of chronic forms of hepatitis B are reported in Belarus annually. Compulsory vaccination, introduced to the country in the mid-90s, produces a certain positive effect on reducing the number of HBV cases. However, around 75,000 patients with chronic hepatitis B are estimated to live in the country at the moment. Our previous study showed that subgenotype D2 (55.8%) was dominant in Belarus, followed by D3 (18,3%), D1 (11,6%) and A2 (11,6%) [18]. The conducted studies show that the structure of genotypes/subgenotypes in the country has actually remained unchanged. Genotype D still dominates in the country and accounts for 81.7% of all analyzed cases. Genotype D was found to be represented by subgenotypes D1 (145/20.1%), D2 (371/51.4%), D3 (201/27, 8%) and D4 (5/0.7%); A – A2; C – C2; B – B4; recombinant forms C2/ D and A2/ С2 were detected. Possible causes of the infection in patients were determined i.e. they occurred due to the circulation of “domestic” variants of the virus and/or the introduction of the virus from outside. The findings obtained show that the epidemic transmission of viral hepatitis B in the country is maintained mainly due to the circulation of local variants of the virus. It is confirmed by the examples of subgenotypes A2, D1, D2 and D3, where the largest clusters were formed by samples from Belarus. However, it should be noted that new variants of HBV are constantly being introduced into the country. The main routes of introduction of A2 subgenotype are Western Europe countries and even the USA, Cuba, Brazil; for D1 - Pakistan, India, Iran, Bangladesh, Italy, Holland and Russia; D2 - Estonia, USA, India, Spain, Russia, Cameroon, Latvia, Iran and Belgium; D3 - Poland, Russia, Rwanda, Estonia and Albania. All samples of D4 subgenotype were clustered with sequences from Cuba and Haiti, C1 and C2 – from China, and B4 – from Vietnam. Two recombinant forms C2/D and A2/D were identified in the residents of our country. Comparing the distribution of different HBV genotypes in neighboring countries and even in different regions of the same country, we may conclude that they vary. For example, genotype D significantly dominates in the European part of Russian Federation, while, for example, in Yakutia, genotype A occupies one of the leading positions (36.4% - A and 58,6% - D) [19]. In Asia, particularly in its eastern part, genotypes B and C dominate, and the C2 subgenotype is epidemic in China [20, 21]. In Western Europe, in particular in Portugal HBV genotype A (HBV/A) was the most prevalent genotype (41.5%), followed by D [HBV/D; (33.8%)], and E [HBV/E; (24.6%)]. Subgenotypes A1 (HBV/A1) and D4 (HBV/D4) were almost equally prevalent with 23.1% and 22.3%, respectively, followed by A2 (HBV/A2) with 16.2% and D3 (HBV/D3) with 11.5% [22].
The Republic of Belarus is geographically located in the center of Europe. The residents of the country are actively travelling and many people from other countries relocate to Belarus for work and study. This fact can easily explain the introduction of the new variants of HBV to the Republic.
The results of this study contribute valuable information of the molecular genetic characteristics of HBV, prevailing in the Republic of Belarus. Possible routes of the introduction of the new variants of HBV have been identified. The main mutations in the P region of the virus genome leading to resistance to antiviral drugs have detected. Identification of the mutations at positions 181T, 204V, 204I, which determine resistance not only to Lamivudine (Zefffix), but also to Adefovir (Hepsera) and Telbivudine (Tyzeka, Sebivo), made it possible to use Tenofovir DF for treatment of the patients. The compensatory mutations 80V, 173L, 80I have been found in the complex with the mutations leading to virus resistance to antiviral drugs.
Conclusions
To summarize, should be noted that the use of the methods of molecular epidemiology solve many issues for both: epidemiologists and infectious diseases specialists. This knowledge is essential to plan preventive measures for the directions and routes of its’ spread in the country by determining the genotypes/subgenotypes of the hepatitis B virus. The quality of living of our patients could be improved by prescribing new treatment regimens, based on the resistance mutations identification.
REFERENCES
- Thun M.J., DeLancey J.O., Center M.M., Jemal A., Ward E.M. The global burden of cancer: Priorities for prevention. Carcinogenesis. 2010;31:100–110.
- Philips C.A., Ahamed R., Abduljaleel J.K., Rajesh S., Augustine P. Critical Updates on Chronic Hepatitis B Virus Infection in 2021. Cureus. 2021;13:e19152.
- Kao J.-H., Chen P.-J., Lai M.-Y., Chen D.-S. Hepatitis B genotypes correlate with clinical outcomes in patients with chronic hepatitis B. Gastroenterology. 2000;118:554–559.
- Aghakhani A., Hamkar R., Zamani N., Eslamifar A., Banifazl M., Saadat A., Sofian M., Adibi L., Irani N., Mehryar M. Hepatitis B virus genotype in Iranian patients with hepatocellular carcinoma. Int. J. Infect. Dis. 2009;13:685–689.
- Beck J., Nassal M. Hepatitis B virus replication. World J. Gastroenterol. 2007;13:48–64.
- Pollicino T., Cacciola I., Saffioti F., Raimondo G. Hepatitis B virus PreS/S gene variants: Pathobiology and clinical implications. J. Hepatol. 2014;61:408–417.
- Ie S.I., Thedja M.D., Roni M., Muljono D.H. Prediction of conformational changes by single mutation in the hepatitis B virus surface antigen (HBsAg) identified in HBsAg-negative blood donors. Virol. J. 2010;7:326.
- Lin C.-L., Kao J.-H. The clinical implications of hepatitis B virus genotype: Recent advances. J. Gastroenterol. Hepatol. 2011;26((Suppl. 1)):123–130.
- Zhu S.S., Zhang H.F., Wang H.B., Dong Y., Chen D., Jia W., Gan Y., Chen J. Relation of viral genotypes to clinical features in children with chronic hepatitis B. Chin. J. Exp. Clin. Virol. 2008;22:192–194.
- Shamseer L., Moher D., Clarke M., Ghersi D., Liberati A., Petticrew M., Shekelle P., Stewart L.A. Preferred reporting items for systematic review and meta-analysis protocols (PRISMA-P) 2015: Elaboration and explanation. BMJ. 2015;349:g7647.
- George B.J., Aban I.B. An application of meta-analysis based on DerSimonian and Laird method. J. Nucl. Cardiol. 2016;23:690–692.
- Fletcher J. What is heterogeneity and is it important? BMJ. 2007;334:94–96.
- Higgins J.P.T., Thompson S.G., Deeks J.J., Altman D.G. Measuring inconsistency in meta-analyses. BMJ. 2003;327:557–560.
- Wallace B.C., Dahabreh I.J., Trikalinos T.A., Lau J., Trow P., Schmid C.H. Closing the Gap between Methodologists and End-Users: R as a Computational Back-End. J. Stat. Softw. 2012;49:2–15.
- Munn Z., MClinSc S.M., Lisy K., Riitano D., Tufanaru C. Methodological guidance for systematic reviews of observational epidemiological studies reporting prevalence and cumulative incidence data. Int. J. Evid. Based Healthc. 2015;13:147–153.
- Yin Y, He K, Wu B, Xu M, Du L, Liu W, Liao P, Liu Y, He M. /Heliyon - 2019 Oct 23;5(10):e02556.
- Serfaty L., Thabut D., Zoulim F., Andreani T., Eres O. C., Carbonell N., Loria A., and Poupon R. / Hepatology – 2001 – V.34 – N.- 3 – P.573-577.
- Olinger C.M., Lazouskaya N.V., Eremin V.F., Muller C.P. Multiple genotypes of hepatitis B and C viruses in Belarus: similarities with Russia and European influences. Clin Microbiol Infect – 2008; 14:575-581.
- Anastasia A Karlsen, Karen K Kyuregyan, Olga V Isaeva, Vera S Kichatova, Fedor A Asadi Mobarkhan, Lyudmila V Bezuglova, Irina G Netesova, Victor A Manuylov, Andrey A Pochtovyi, Vladimir A Gushchin, Snezhana S Sleptsova, Margarita E Ignateva, Mikhail I Mikhailov Different evolutionary dynamics of hepatitis B virus genotypes A and D, and hepatitis D virus genotypes 1 and 2 in an endemic area of Yakutia, Russia // /BMC Infect Dis – 2022; 12;22:452
- Kizito Eneye Bello, Tuan Nur Akmalina Mat Jusoh, Ahmad Adebayo Irekeola, Norhidayah Abu, Nur Amalin Zahirah Mohd Amin, Nazri Mustaffa, Rafidah Hanim Shueb/ A Recent Prevalence of Hepatitis B Virus (HBV) Genotypes and Subtypes in Asia: A Systematic Review and Meta-Analysis // Healthcare (Basel) – 2023; 1;11(7):1011.
- Bing Sun, Aida Andrades Valtueña, Arthur Kocher, Shizhu Gao, Chunxiang Li, Shuang Fu, Fan Zhang, Pengcheng Ma, Xuan Yang, Yulan Qiu, Quanchao Zhang, Jian Ma, Shan Chen, Xiaoming Xiao, Sodnomjamts Damchaabadgar, Fajun Li, Alexey Kovalev, Chunbai Hu, Xianglong Chen, Lixin Wang, Wenying Li, Yawei Zhou, Hong Zhu Johannes Krause, Alexander Herbig, Yinqiu Cui/ Origin and dispersal history of Hepatitis B virus in Eastern Eurasia // Nat Commun, 2024; 5;15:2951.
- Rute Marcelino, Ifeanyi Jude Ezeonwumelu, André Janeiro, Paula Mimoso, Sónia Matos, Veronica Briz, Victor Pimentel, Marta Pingarilho, Rui Tato Marinho, José Maria Marcelino, Nuno Taveira, Ana Abecasis / . Phylogeography of hepatitis B virus: The role of Portugal in the early dissemination of HBVworldwide // PLoS One. 2022; 17(12):
Acknowledgments
We would like to show our gratitude to the Abbott Transfusion Medicine Medical affairs team for financial support of the publication.
Abstract. Annually 800-900 new cases of hepatitis B chronic forms are registered in Belarus. Compulsory vaccination introduced in the mid-90s certainly produces a positive effect on reducing the number of HBV cases. About 75,000 patients with chronic hepatitis B living in the country are estimated to live in the country at the moment. The main goal of this work was to determine the genotypes/subgenotypes of the hepatitis B virus and establish their role in maintaining the epidemiological process on the territory of Belarus. Serological (CMIA, ELISA), molecular biological (PCR, sequencing) and bioinformatic (phylogenetic analysis) methods have been used. As studies have shown, out of all 884 sequenced and analyzed samples, 722 (81.7%) of them have genotype D; 156 (17.7%) - genotype A; 3 (0.3%) - genotype C and the only 1 (0.1%) – genotype B. For two cases (0.2%) a recombinant form of the hepatitis B virus was detected. The epidemic process of hepatitis B in the country is supported mainly by the circulation of “local” variants of the virus. At the same time, there are occasional introductions of new variants from outside the country.
Keywords. HBV, genotypes/subgenotypes, sequencing, phylogenetic analysis.